# The Labyrinthine Landscape of APP Processing: State of the Art and Possible Novel Soluble APP-Related Molecular Players in Traumatic Brain Injury and Neurodegeneration

**DOI:** 10.3390/ijms24076639

**Published:** 2023-04-02

**Authors:** Mirco Masi, Fabrizio Biundo, André Fiou, Marco Racchi, Alessia Pascale, Erica Buoso

**Affiliations:** 1Computational and Chemical Biology, Italian Institute of Technology, Via Morego 30, 16163 Genova, Italy; masi.mirco1994@gmail.com; 2Department of Developmental and Molecular Biology, Albert Einstein College of Medicine, 1300 Morris Park Ave, Bronx, NY 10461, USA; fabrizio.biundo@einsteinmed.edu; 3Department of Drug Sciences, Pharmacology Section, University of Pavia, Via Taramelli 12/14, 27100 Pavia, Italyalessia.pascale@unipv.it (A.P.); buoso.erica@gmail.com (E.B.); 4Department of Pharmacology and Experimental Therapeutics, Boston University School of Medicine, Boston, MA 02118, USA

**Keywords:** *BACE1*, *TSPAN3*, *VEGF*, *ISM2*, ELAV, secretase, PKC, RACK1

## Abstract

Amyloid Precursor Protein (APP) and its cleavage processes have been widely investigated in the past, in particular in the context of Alzheimer’s Disease (AD). Evidence of an increased expression of APP and its amyloidogenic-related cleavage enzymes, β-secretase 1 (*BACE1*) and γ-secretase, at the hit axon terminals following Traumatic Brain Injury (TBI), firstly suggested a correlation between TBI and AD. Indeed, mild and severe TBI have been recognised as influential risk factors for different neurodegenerative diseases, including AD. In the present work, we describe the state of the art of APP proteolytic processing, underlining the different roles of its cleavage fragments in both physiological and pathological contexts. Considering the neuroprotective role of the soluble APP alpha (sAPPα) fragment, we hypothesised that sAPPα could modulate the expression of genes of interest for AD and TBI. Hence, we present preliminary experiments addressing sAPPα-mediated regulation of *BACE1*, Isthmin 2 (*ISM2*), Tetraspanin-3 (*TSPAN3*) and the Vascular Endothelial Growth Factor (*VEGFA*), each discussed from a biological and pharmacological point of view in AD and TBI. We finally propose a neuroprotective interaction network, in which the Receptor for Activated C Kinase 1 (RACK1) and the signalling cascade of PKCβII/nELAV/VEGF play hub roles, suggesting that vasculogenic-targeting therapies could be a feasible approach for vascular-related brain injuries typical of AD and TBI.

## 1. Introduction

Alzheimer’s Disease (AD) is a chronic neurodegenerative disease characterised by a progressive impairment of cognitive functions ultimately resulting in dementia. Many hypotheses for AD development have been proposed, including the formation of twisted fibres of Tau proteins inside neurons called Neurofibrillary Tangles (NFTs), ribosomal impairment and the RNA-binding protein cascade hypothesis [1,2]. However, one of the most accredited theories suggests the abnormal deposition of Amyloid β (Aβ) protein oligomers in neuronal intracellular space as one of the main pathology drivers [3]. Aβ is a 4 kDa peptide derived from the cleavage of its precursor, the Amyloid Precursor Protein (APP), whose role in the pathogenesis and progression of AD has been intensively investigated [4]. Neuropathologic markers like NFTs and senile plaques composed of Aβ aggregates histopathologically characterise the brain tissue of AD patients [5]. In this regard, the AD amyloid hypothesis focuses on the toxic role of the excessive formation of Aβ, which tends to accumulate into extracellular senile plaques, directly responsible for AD pathogenesis. Accordingly, the excessive formation of Aβ derives either from an increased production of Aβ after APP processing or from its reduced elimination [3]. Mutations in *APP* gene, located on chromosome 21, have played an important role to understand AD aetiology, although APP mutations accounted only for a small fraction of all AD cases. Further studies reported mutations also in different APP processing-related genes, including Beta-site APP Cleaving Enzyme 1 (*BACE1*, that codes for β-secretase 1), the structural components of the γ-secretase complex *PSEN1* and *PSEN2* (located on chromosome 14 and 1 and coding for Presenilin 1 (PS1) and PS2, respectively) and other genes belonging to the γ-secretase complex [3]. In addition, the ε4 polymorphism in the *APOE* gene (coding for apolipoprotein E, which modulates Aβ oligomerisation into fibrils leading to senile plaques formation) is associated with an increased risk of late-onset AD [6] and represents a major susceptibility risk factor for AD [7].

Observational epidemiological studies suggested a series of additional environmental influences that can be either protective or risk factors for AD. Among the protective factors, anti-inflammatory and anti-oxidant influences associated with increased brain neurogenesis and Brain-Derived Neurotrophic Factor (BDNF) production promote a healthier brain aging [8,9], while a history of cancer appears to be beneficial at reducing the likelihood of AD development [10]. Environmental risk factors for AD include depression [11,12], cardiovascular and metabolic status and history of head injury [13]. In this regard, Traumatic Brain Injury (TBI) has been suggested to trigger a deleterious cascade of secondary damage, leading to neuroinflammation, persistent neurological and cognitive impairment, and ultimately dementia [14,15]. Both mild and severe TBI are influential risk factors for different delayed-onset neurodegenerative diseases, including AD [16,17]. Following TBI with axonal transection, APP, β-secretase 1 and γ-secretase (the enzymes that contribute to APP cleavage towards Aβ formation) show an increased expression particularly prominent at the hit axon terminals [18,19,20]. This leads to an increased Aβ production and deposition at the axon bulbs and strengthens the correlation between TBI and increased risk for AD.

APP and the implications of its processing are important not only for neurodegeneration, particularly for AD, but also for TBI. Hence, the aim of this work is to present the state of the art of APP proteolytic processing, highlighting the different roles of its cleavage fragments in both physiological and pathological contexts, with a particular focus on the soluble APP fragments. In addition, we aim to discuss specific genes of interest for AD and TBI, and regulated by soluble APP peptides emerged from our preliminary in vitro data here presented from a biological and pharmacological point of view.

## 2. Amyloid Precursor Protein: Structure, Expression and Processing

### 2.1. APP Structure, Expression, Trafficking and Modification

#### 2.1.1. APP Structure and Expression

The human *APP* gene, which is mapped to chromosome 21q21.3, spans 290,586 bp and consists of 18 exons. As a result of alternative splicing, the generation of multiple *APP* mRNA isoforms allows the production of different APP proteins with a number of amino acids ranging from 365 to 770 residues (APP365–APP770) [4] (Figure 1a). The highly conserved APP family consists of APP, the homologous APP-Like Protein 1 (APLP1) and APLP2, which are single-pass integral membrane proteins that feature a bulky N-terminal extracellular domain, an intramembrane domain and a small, intracellular C-terminal tail characterised by a sequence identity highly conserved among the three APP family members. However, APP is the only member of the family characterised by the presence of an Aβ domain [21]. The glycosylated N-terminal portion is relatively conserved among the different APP family members and the different APP isoforms and contains: the Signal Peptide (SP) sequence, the cysteine-rich globular domain (E1), the Extension Domain (ED), the Acidic Domain (AcD), a helix-rich domain (E2) and part of the Aβ domain, [4]. The E1 domain features a Heparin-Binding Site (HBS) (that confers APP its neuroprotective effects [22]) and a Metal-Binding Motif (MBM) with Cu^+^ and Zn^2+^ binding sites (ZnBS and CuBS, respectively). Specific splicing variants feature additional domains, e.g., the Kunitz Protease Inhibitor (KPI) after the AcD domain and the OX-2 antigen domain after the KPI domain, while others lack these particular regions [4]. The E2 domain features six α-helices that form a coiled-coil structure, an HBS and a Collagen Binding Site (CBS). Importantly, the short cytoplasmic C-terminal domain contains the conserved YENPTY motif important for the protein–protein interactions (Figure 1b). Both APP N-terminal and C-terminal portions are involved in different cellular mechanisms (reviewed in [23]) and appear to participate in several signalling pathways as APP full-length and cleavage fragments. In the Central Nervous System (CNS), APP works as a cell surface receptor and is correlated with neurite growth, neuronal adhesion and axonogenesis [24]. However, increasing evidence suggests that both APP and its cleavage fragments play important roles also in peripheral tissues, where their abnormal expression, location and cleavage have been linked to the development of metabolic diseases (reviewed in [23]). Regarding APP isoforms, different splicing variants exhibit a tissue-specific expression and, among the brain-specific APP splicing variants, APP695 is the most abundant isoform expressed in the brain [25], although APP751 and APP770 are the main coding proteins of Aβ peptide [4]. Among the different APP family members, APLP1 expression is restricted to neurons and, while APLP2 and APP are highly brain-enriched and their expression has been found also in several peripheral tissues. While APP and APLP2 appear to be functionally redundant, the solely CNS-expressed APLP1 may have distinct roles as part of the synaptic network (reviewed in [26]). Indeed, recent findings point at APLP1 functions in neuronal morphology [27], synaptic plasticity [28,29], dendritic spine maintenance [30] and even as a possible biomarker of AD progression [31].

#### 2.1.2. APP Trafficking and Post-Translational Modifications (PTMs)

After its synthesis, APP is firstly translocated to the Endoplasmic Reticulum (ER) upon the removal of its SP sequence, then it transits the ER/ER-Golgi Intermediate Compartment (ERGIC) to locate into the Golgi apparatus and the Trans-Golgi Network (TGN). Although the majority of APP localises in the Golgi apparatus and TGN, a small fraction of the nascent APP is translocated to the plasma membrane, where either α-secretase mediates its non-amyloidogenic cleavage or its C-terminal YENPTY motif mediates its internalisation into endosomes [32]. Internalised APP then traffics through the endo-lysosome pathway (mostly for APP degradation or, to a less extent, for its recycling to the plasma membrane or TGN), secretory pathway and recycling pathway. APP trafficking is strictly correlated with its processing, since α-secretase mainly resides in the plasma membrane, while β-secretase 1 locates in endosomes and lysosomes [32]. In the secretory pathway, APP can undergo different PTMs that influence its residency and trafficking, ultimately affecting the production of the different APP fragments. In this regard, each of APP PTMs here presented have different effects on Aβ generation (reviewed in [32]) and abnormal APP PTMs and alterations of its trafficking have been reported in AD patients [32].

Glycosylation and phosphorylation

Upon its translocation into the ER, APP is N-glycosylated at N467 and N496 by the Oligosaccharyl Transferase (OST) complex forming immature APP, while APP O-glycosylation at multiple sites (T291, T292, T353, T576, S606, S611, T616, T634, T635, S662 and S680 found in cell cultures and human Cerebrospinal Fluid (CSF)) occurs in Golgi apparatus to allow APP maturation and its localisation in TGN and the plasma membrane [32]. APP PTMs and the interplay between APP classical O-GalNAcylation and O-GlcNAcylation (i.e., the addition of a single β-N-acetylglucosamine) at serine and threonine residues is pivotal for APP trafficking. Its N-glycosylation is essential for APP sorting from the Golgi apparatus to the plasma membrane as well as for its transport to the axonal synaptic membrane [32]. APP O-GlcNAcylation favours its trafficking from the TGN to the plasma membrane while inhibiting its endocytosis. Indeed, alterations of this APP O-glycosylation is crucial for the regulation of APP processing and Aβ production [33].

APP can undergo phosphorylation at two sites in the ectodomain (S198, S206) and eight sites in the cytoplasmic domain (Y653, Y682, Y687, S655, S675, T654, T668, T686) [32]. These PTMs are mainly catalysed by Protein Kinase C (PKC) (on S655, mainly detected in the mature APP), Ca^2+^/Calmodulin-dependent Protein Kinase II and APP kinase I (on S655 and T654), Glycogen Synthase Kinase 3 beta (GSK-3β), Cyclin-Dependent Kinase 1 (CDK1), CDK5, Stress-Activated Protein Kinase 1 beta (SAPK1β), Dual-specificity Tyrosine phosphorylation-Regulated Kinase 1A (DYRK1A) and c-Jun N-terminal protein Kinase (JNK) (on T668 mainly detected in immature APP and occurring in the ER) [32]. APP phosphorylation is required for its trafficking as demonstrated by mutagenesis experiments [32].

Palmitoylation, ubiquitination, SUMOylation and sulphation

The two palmitoyl acyltransferases DHHC-7 and DHHC-21 catalyse APP palmitoylation at C186 and C187 in the ER. APP palmitoylation is pivotal for the regulation of its trafficking, maturation, localisation to lipid rafts and its interactions with other proteins as demonstrated by double mutagenesis experiments on the involved cysteine residues [34]. Regarding ubiquitination, different ubiquitin-activating enzymes (E1), ubiquitin-conjugating enzymes (E2) and ubiquitin ligases (E3) act in concert to catalyse APP ubiquitination at residues K649–651 and K688 located in its ectodomain to regulate maturation, degradation and protein–protein interactions. This process is required for APP sorting and trafficking, especially to the endosomal compartment, as observed in hippocampal neurons [35]. Conversely, APP ubiquitination mediated by F-Box/LRR-repeat protein 2 (FBXL2) inhibits its endocytosis, increases its exposure to the plasma membrane and decreases its presence in lipid rafts [36]. Similarly to ubiquitination, SUMOylation (i.e., the covalent addition of Small Ubiquitin-like Modifier (SUMO) SUMO-1, -2 and -3) is catalysed by SUMO E1, E2 and E3 at APP K587 and K595 to regulate its functions [37]. Finally, APP sulphation (common PTM for cell surface proteins) at Y217 and Y262 residue occurs in the late Golgi compartment and has been hypothesised to be implicated in APP trafficking and degradation, although APP sulphation sites and functions still need to be completely elucidated [32].

### 2.2. APP Processing

As previously mentioned, APP can undergo different cleavage mechanisms that differentially produce several smaller peptides. Notably, APP cleavage processing involves a canonical and a non-canonical pathway. APP canonical cleavage processing includes two different proteolytic mechanisms, i.e., the non-amyloidogenic and the amyloidogenic pathways, that differently cleave APP via α-, β-, and γ-secretases and release different proteolytic products. Rather than a single α-secretase, several A Disintegrin And Metalloprotease (ADAM) family members, in particular ADAM9, ADAM10 and ADAM17 (also known as Tumour necrosis factor α-Converting Enzyme, TACE) are involved in APP cleavage [21,38]. In addition, unlike α-secretase, β-secretase 1 is a single transmembrane aspartyl protease. Finally, γ-secretase is a high molecular weight complex that consists of PS1 and/or PS2, Presenilin Enhancer 2 (PEN2), Anterior Pharynx defective 1 (APH1) and Nicastrin (NCSTN, also known as APH2) [39]. On the other hand, APP non-canonical cleavage includes different soluble and membrane-bound secretases whose mechanisms and their actual contribution to AD are still under investigation.

#### 2.2.1. APP Canonical Cleavage: Amyloidogenic and Non-Amyloidogenic Pathways

In the amyloidogenic pathway, APP is firstly cut by β-secretase 1 in endosomes at the N-terminal side of the Aβ sequence (termed β-site, M671-D672). This produces an externally released, N-terminally truncated APP form termed soluble APPβ (sAPPβ) and the membrane-associated C-terminal fragment (C99 or β-CTF), which remains associated to the endosomal system (Figure 2a, right green panel). A second, less prominent β-secretase 1 cleavage site (termed β′-site, Y681-E682) located 10 residues to the C-terminus of APP leads to the generation of β′-CTF (or C89), whose further cleavage results in the production of the N-terminally truncated Aβ_11–X_ [40]. sAPPβ can undergo a further cleavage at APP286 residue to produce N-APP [41], a 35 kDa peptide reported to bind death receptor 6 (DR6, also known as Tumour Necrosis Factor Receptor Superfamily member 21, TNFRSF21) thus triggering apoptosis and axon pruning, but also neuronal death and possibly AD development [42,43]. The β-CTF fragment is then cleaved into two additional peptides, i.e., APP Intracellular Domain (AICD) and Aβ peptides, by the γ-secretase complex. AICD_50_ (composed of 50 residues) is the dominant fragment, although other species (i.e., AICD_48_, AICD_51_ and AICD_53_) have also been identified and demonstrated to result from additional γ-secretase-mediated cleavages. In contrast, in the non-amyloidogenic pathway, APP is firstly cut by α-secretase between K687-L688 (termed α-site) in the plasma membrane, cleaving APP within the Aβ sequence and thereby preventing Aβ formation. This proteolytic cleavage produces soluble APPα (sAPPα), a secreted peptide released into the extracellular space, and the membrane-associated C-terminal fragment (C83 or α-CTF) [44]. From this remaining peptide, two additional peptides, i.e., AICD and the 3 kDa product p3, are generated after its cytoplasmatic cleavage by the γ-secretase complex (Figure 2a, left blue panel). Even though they share a similar peptide sequence, AICD produced in these two proteolytic pathways appear to have distinct functions. AICD produced through the non-amyloidogenic pathway undergo a rapid cytoplasmatic degradation through the endo-lysosomal system [45] and by the Insulin-Degrading Enzyme (IDE, a large zinc-binding protease of the M16 metalloprotease family) [46,47]. In contrast, AICD generated via the amyloidogenic pathway bind the Fe65 adaptor protein and translocate into the nucleus, where they associate with Tip60 to form the ATF complex [48]. The ATF complex acts as a transcription factor and regulates the expression of different APP-related genes, including *APP* itself (its own precursor), *BACE1*, *GSK3B* (coding for GSK-3β) and *MME* (coding for the Aβ-degrading enzyme Neprilysin, also known as Membrane Metallo-Endopeptidase (MME), Neutral Endopeptidase (NEP), cluster of differentiation 10 (CD10), and Common Acute Lymphoblastic Leukaemia Antigen (CALLA)) among the others [49,50,51].

The two APP homologues APLP1 and APLP2 are similarly processed by the same secretases, producing the intracellular fragments APLP-Intracellular Domain 1 (ALID1) and ALID2, respectively [52]. Similarly to AICD, ALID1 and ALID2 have been proposed to play a functional role as transcriptional regulators [53], although the validity of these speculations is still controversial due to the lack of adequate animal models.

The γ-secretase complex can operate different proteolytic cleavages in the CTFs produced by both amyloidogenic and non-amyloidogenic pathways. The proteolytic cleavage operated by γ-secretase can be further separated in the sequentially occurring γ-, ζ-, and ε-cleavage sites (five γ-sites, two ζ-sites and two ε-sites) [54] (Figure 1b). This is particularly important for the generation of Aβ species, since the diverse possible cleavages exerted by γ-secretase result in the production of different Aβ peptides. After β-secretase 1-mediated APP cleavage, β-CTFs are processed through ε-cleavage, either producing Aβ_49_ and AICD_50–99_ or Aβ_48_ and AICD_49–99_. Within the Aβ_40_ product line, Aβ_49_ is cleaved at the ζ-site to Aβ_46_ and at γ-sites to Aβ_43_, Aβ_40_ and Aβ_37_ via a tripeptide trimming. On the other hand, within the Aβ_42_ product line, Aβ_48_ is cleaved at the ζ-site to Aβ_45_ and at γ-sites to Aβ_42_ and Aβ_38_ via a tetrapeptide trimming [55]. A balanced generation and elimination of Aβ peptide products occurs in the healthy brain and the length of Aβ species found in CSF ranges from 37 to 43 amino acids [56]. However, in AD brain, Aβ peptides are more prone to aggregation into toxic amyloid oligomers—that eventually evolve in protofibrils and fibrils—via different assembly processes, affecting neuronal function and synaptic activity ultimately leading to synapse loss and impaired cerebral capillary blood flow [57]. The production ratio Aβ_42_/Aβ_40_ is around 1 to 9 but, due to its enhanced hydrophobicity, Aβ_42_ has a stronger aggregation ability and is more toxic than Aβ_40_. Indeed, Aβ_42_ is the major amyloid plaques component, although Aβ_43_ was also reported in the human AD brain amyloid deposition [23,56].

#### 2.2.2. APP Non-Canonical Cleavage

Although its canonical cleavage has been thoroughly investigated in the past, the complete APP processing involves an increasing number of additional secretases—globally referred to as APP non-canonical cleavage—able to proteolytically cleave APP in both the ectodomain and the intracellular domain, recently identified also in vivo [58].

δ-secretase

δ-secretase (also known as Asparagine Endopeptidase (AEP) and Legumain) is a soluble, pH-controlled lysosomal cysteine protease, previously linked to AD due to its ability to proteolytically cleave Tau [59] and recently reported to participate in APP ectodomain processing [60]. δ-secretase has been shown to cut APP between N373-E374 and N585-I586 both in vitro and in vivo, producing three soluble fragments (sAPP_1–373_, sAPP_1–585_ and sAPP_374–585_) and the membrane-bound C-terminal fragment (δ-CTF), which is then processed by β- and γ-secretases, resulting in Aβ and AICD release [60] (Figure 2b). Increased δ-secretase expression and activity were reported in aged mice and in the brains of AD patients. On the other hand, its knock-out (KO) resulted in decreased Aβ production, dendritic spine and synapse loss, as well as behavioural impairments, protecting against memory deficits in two different AD mice models (i.e., 5 × FAD and APP/PS-1) [60]. In addition, the δ-secretase-generated APP C586–695 fragment has been shown to directly bind the inflammation-related CCAAT/Enhancer Binding Protein beta (C/EBPβ) transcription factor, eliciting its nuclear translocation and enhancing its transcriptional activity. This resulted in the induced transcription and expression of *APP*, *MAPT* (coding for the microtubule-associated protein Tau), *LGMN* (coding for δ-secretase) and inflammatory cytokines, escalating AD-related gene expression and pathogenesis and resulting in AD pathology and cognitive disorder [61]. These results suggest a role for δ-secretase in AD, although further studies are required to confirm and elucidate its contribution to AD and to consider δ-secretase as a potential drug target.

η-secretase

η-secretase (also known as Membrane-type 5-Matrix Metalloproteinase, MT5-MMP, and MMP-24) is a glycosylated transmembrane proteinase and Zn^2+^-dependent MMP, intracellularly activated by the Ca^2+^-dependent proprotein convertase furin and primarily expressed in neural cells [62]. η-secretase has been shown to contribute to both physiologic and pathologic processes in the nervous system and its co-localisation with senile plaques suggested its possible involvement in AD pathogenesis [62]. η-secretase cleaves APP between N504-M505, producing a N-terminal soluble 95 kDa fragment (sAPP95 or sAPPη) and the membrane-bound C-terminal fragment (η-CTF or C191) [63,64]. η-CTF is then cut by α- or β-secretase, releasing the soluble fragments Aη-α or Aη-β, respectively, while the remaining CTF is then cleaved by the γ-secretase complex, producing p3 and AICD or Aβ and AICD, respectively [63,64] (Figure 2c). In addition to proteolytically cleave APP and release Aβ both in vitro and in vivo [65,66], η-secretase has been shown to contribute to the production of proteolytic fragments able to induce synaptic dysfunction [63,64]. Notably, η-secretase KO in 5xFAD mice led to a significant reduction of Aβ plaque deposition in the brain, along with preserved hippocampal function and improved spatial memory [63]. These observations indicate a potentially important role for η-secretase in AD context, although further investigations to confirm its pathophysiologic implications are required.

Meprin-β

Meprin-β is a Zn^2+^ metalloprotease that acts as β-secretase 1 and competes with it [67], proteolytically cleaving within APP β-site M596-D597 as well as in the adjacent sites (D597-A598 and A598-E599) [68]. However, unlike for β-secretase 1, this APP cleavage takes place at the cell surface, indicating a direct competition with α-secretase in vivo [67]. Meprin-β proteolytic cleavage produces a fragment similar to sAPPβ (sAPPβ*) and two shorter soluble fragments (sAPP_1–380/3_ and sAPP_1–124_), while the remaining membrane-bound CTF (β*-CTF) undergoes a γ-secretase-mediated cleavage releasing Aβ_2–X_ and AICD (Figure 2d). However, Aβ_2–X_ abundance in the brain is severalfold lower compared to Aβ_42_ species [67], indicating that meprin-β possible contribution to AD pathogenesis needs further investigation. Moreover, meprin-β KO resulted in an increased production of sAPPβ, altering the Aβ_2–40_/Aβ_1–40_ ratio [67]. In addition to the β-site, meprin-β can cleave APP in three further sites, between A124-D125 (resulting in the production of the 11 kDa fragment p11), E380-T381 and G383-D384 [69]. Therefore, it has been proposed that meprin-β can process APP by proteolytically cleaving in these different sites depending on APP subcellular localisation (soluble meprin-β releases the N-terminal APP fragments, while membrane-bound meprin-β cleaves at APP β-site) [70].

θ-secretase

θ-secretase (also known as β-secretase 2, encoded by *BACE2*) is a *BACE1* homologue located on chromosome 21q22.2-q22.3. θ-secretase-mediated APP cleavage releases the soluble N-terminal fragment sAPPθ and the membrane-bound C-terminal fragment (θ-CTF or C80), which is further cleaved by the γ-secretase complex, producing AICD and truncated Aβ (Figure 2e). By performing a proteolytic cleavage at APP θ-site F690-F691, downstream of the α-site, θ-secretase cleaves APP within the Aβ domain, abolishing Aβ production [71]. For this reason, it is considered as a possible therapeutic target to prevent the disease progression. Its relevance in AD has been suggested not only for its role as θ-secretase but also as a conditional β-secretase capable of processing APP at β-site, although APP Juxtamembrane Region (JMR) normally inhibits this activity. An increased binding of clusterin protein to JMR and the presence of JMR-disrupting mutations have been found in aged mouse brain, which correlated to an enhanced θ-secretase β-cleavage during aging. In this regard, both clusterin-JMR binding and JMR mutations prevent APP θ-cleavage, favouring β-cleavage of nascent APP and worsening AD symptoms [72]. Indeed, θ-secretase has been reported to be dysregulated in AD and this dysregulation has been hypothesised to be mediated by proteasomal and lysosomal impairments observed in AD [73]. However, since θ-secretase regulation has been less investigated compared to β-secretase 1, further studies are needed to elucidate its contribution to AD as well as putative drug target.

Other secretases

The identification of additional Aβ peptides—both N-terminally truncated (e.g., Aβ_5–X_) and N-terminally extended [74,75]—and APP N-terminal fragments (e.g., N-APP_18–286_ fragment) [76,77] in human CSF suggests that APP could be proteolytically cleaved by additional secretases. Different caspases—namely caspase-3, caspase-6 and caspase-8 (reviewed in [58])—can mediate APP C-terminal domain cleavage at D664. This cleavage releases a membrane bound N-terminal fragment (APP-Ncas) and the C-terminal 31 residues-long fragment (APP-Ccas or C31) in the cytoplasm [64], which plays a role in neuronal apoptosis [58]. Notably, after α- or β-secretase 1 cut, the resulting CTF undergoes a combined cleavage by γ-secretase and caspase, releasing p3 or Aβ, respectively, in the extracellular space and the small intracellular peptide JCasp (Figure 2f), which interacts with and sequesters proteins involved in the vesicle-release machinery [78]. Moreover, the Rhomboid-like protein-4 (RHBDL4) is mainly located in the ER and belongs to the five-member family of intramembrane proteinases rhomboids (the secretory pathway-located RHBDL1-4 and the mitochondrial Presenilin Associated Rhomboid Like (PARL)). RHBDL4 cleaves APP ectodomain at multiple sites, generating ~70 kDa different N-terminal and C-terminal fragments with unknown functions. RHBDL4-mediated APP cleavage is negatively regulated by cholesterol and decreases Aβ_38_, Aβ_40_ and Aβ_42_ levels, although these in vitro observations still need to be validated for their relevance regarding AD pathogenesis [79]. Finally, Cathepsin B is a lysosomal cysteine protease proposed as a putative β-secretase and whose role in Aβ pathology is still debated due to conflicting evidence [80,81,82]. Cathepsin B cleaves APP at the A673-E674 site, producing the Aβ peptides Aβ_3–X_ or Aβ_11–X_, whose N-terminal glutamate residue undergoes a glutaminyl cyclase-mediated cyclisation to release N-terminally truncated pyro-glutamylated Aβ peptides (pE-Aβ) [83].

### 2.3. APP and Its Processing in TBI

An increased expression of APP, β-secretase 1 and γ-secretase components, accompanied by increased Aβ production and deposition, were observed at the hit axonal terminals after both mild and severe TBI [18,19,20], hinting at a possible correlation between TBI history and AD development. Epidemiological evidence highlighted that even remote TBI history may anticipate AD onset trajectory, with greater Aβ deposition accompanied by cortical thinning [84]. Indeed, an elevated accumulation of Aβ peptides—in particular the Aβ_42_ species—in peripheral blood was reported among patients with a history of TBI with impaired cognition [85]. Moreover, Aβ persistency in the blood of both mild and severe TBI patients even months after the injury incident [86] clearly indicates a correlation between TBI, Aβ production and presence of symptoms. In this regard, TBI activates C/EBPβ transcription factor, leading to increased δ-secretase expression and activity, mediating AD pathogenesis by promoting Aβ production as well as Tau hyperphosphorylation (Figure 3a). This results in neurotoxicity and neuroinflammation [87,88], which is also exacerbated in the presence of a defective BDNF and TrkB neurotrophic signalling [89]. KO of *CTSB* gene that codes for Cathepsin B—involved in the production of Aβ peptides relevant for AD—resulted in the amelioration of brain dysfunctions in both AD and TBI contexts, improving behavioural deficits and neuropathology in both diseases [90]. Moreover, caspase-3-mediated APP processing has been observed in axons undergoing traumatic axonal injury after TBI, co-localising with Aβ formation and suggesting that non-canonical APP processing pathways may have an important role in TBI and future AD development [91]. Although the precise mechanism that links TBI with APP alterations and Aβ production has still to be completely elucidated, an epigenetic and a physical components have been proposed. In this regard, TBI has been reported to induce a differential CpG methylation of different genes related to neurodegeneration, among which *APP* and *MAPT* were top differentially methylated CpG sites [92]. As for the physical component, the inertial loading stress to the head causes a dynamic mechanical shearing within the brain, leading to a deformation of the brain tissue that damages long-tract structures like axons and blood vessels [93]. Notably, APP is transported through a fast axoplasmic transport and accumulates in proximity of the axonal injury site [93]. Hence, the TBI-induced distortion of the axonal cytoskeleton impairs normal transport and triggers APP cleavage in the injured region, disrupting APP axonal transport and leading to Aβ generation and alteration of neuronal homeostasis [94] (Figure 3b).

## 3. Physiological and Pathological Roles of APP and Its Cleavage Fragments

Although it has been reported to act as a cell surface receptor to facilitate cell adhesion and regulate synapse formation and cell division, APP has been shown to exhibit both neuroprotective and neurotoxic effects [4]. However, both in vitro and in vivo models employed to investigate APP processing pathways and their contribution to AD and TBI pathologies rely on its over-expression or down-regulation. For these reasons, addressing the precise physiological roles of non-cleaved APP is rather challenging. In addition, most literature data focused their studies on Aβ or other APP fragments and the possible existence of compensatory mechanisms of APLP1 and APLP2 further complicate its investigation [4]. Despite these experimental caveats, full-length APP and its cleavage fragments have been widely studied in the past decades, thus generating an increasing amount of data that contributes to extricate this complex landscape in both physiological and pathological contexts. Hence, for a more functional presentation of their effects reported in literature, properties of the different proteolytic fragments generated from APP processing pathway here treated, as well as full-length APP functions, are presented in Table 1.

The literature data reported that α-secretase-mediated APP processing occurs via constitutive pathways in response to physiological processes in neuronal circuits and to nervous system injury like TBI, specifically through multiple receptor-mediated activation of signal transduction pathways, among which PKC plays a pivotal role [131,132]. As reported in Table 1, several studies suggested sAPPα involvement in neuroprotective mechanisms, such as synaptic plasticity, synaptogenesis and neurite outgrowth as well as neurotrophic actions [133]. It is well known that the dysregulation of APP processing and metabolism is correlated to neurological disorders not limited to AD, but also Down’s Syndrome (DS), Fragile X Syndrome and Autism, in which increased sAPPα levels contributed to an excessive induction and stimulation of brain growth signals during development [134,135,136]. Conversely, reductions of constitutive and regulated release of sAPPα and decreased sAPPα levels have been reported in the CSF of AD patients [137,138,139]. Therefore, a loss of sAPPα neuroprotective and neurotrophic functions could be involved in the decreased neuronal plasticity and the increased neuronal susceptibility to cellular stress observed in aging and neurodegeneration. Experimental evidence suggests that sAPPα neuroprotective actions are correlated with effects on ion channel function followed by late transcription-dependent events [140], as well as pleiotropic effects on cell survival via the PI3K/Akt/NF-κB pathway as we previously demonstrated [141]. In this context, a deeper understanding of the sAPPα-correlated modulation of genes of interest for both AD and TBI could be significant in terms of basic research and future pharmacologic intervention [142].

## 4. Results

### 4.1. Neuron-Related Differential sAPPα-Mediated Gene Regulation In Vitro

Experiments on sAPPα-treated SH-SH5Y cells—which are widely used as preliminary in vitro models for both AD and TBI preclinical investigations [143,144]—resulted in the activation of MAPK-related pathway (Figure 4a–c) in line with literature data [145], and consequently in the differential modulation of the expression of different genes at mRNA level, as reported in Table 2. Genes of our interest (bold and highlighted in Table 2) were then validated through quantitative PCR (qPCR).

We performed preliminary experiments on four sAPPα-modulated target genes, (i.e., *BACE1*, *TSPAN3*, *VEGFA* and *ISM2*) with potential correlations with sAPPα neuroprotective actions. Then, we discussed these genes in light of sAPPα neuroprotective actions aiming to unravel possible novel cytosolic- and membrane-related molecular players interesting for AD and TBI contexts [141,142]. Other sAPPα modulated genes are presented in Table 3 with their putative or reported roles in AD and TBI.

### 4.2. BACE1

The human *BACE1* gene is located on chromosome 11q23.3 and encodes for β-secretase 1, the main APP degrading enzyme involved in the amyloidogenic pathway. Research on BACE1 has been historically focused on its role in AD development/Aβ production and its neuronal and non-neuronal roles, as well as its state of the art, have been recently and excellently reviewed [160]. *BACE1* is abundantly expressed in brain and pancreatic tissues, although its low expression is also detected in many other cell types and has been investigated in multiple pathological contexts [160]. Together with its homologue *BACE2*, *BACE1* encodes for a type 1 membrane protein part of the membrane-bound aspartyl proteases subfamily [168]. β-secretase 1 is a 501 residues pre-protein, whose structure features five domains: SP, Pro-Peptide (PP), Catalytic Domain (CD), Transmembrane Domain (TM) and the C-Terminal Region (CTR). The SP traffics BACE1 to the ER, where furin cleaves PP to form mature β-secretase 1, while the TM domain localises β-secretase 1 to the late Golgi and its post-transcriptional activation occurs in the trans-Golgi. Subsequently, β-secretase 1 proteolytically cleaves its substrates in endosomes while being membrane-bound [168]. Due to its involvement in APP amyloid cleavage processing, β-secretase 1 has a key role in controlling Aβ production. Noteworthy, sAPPα has been shown to act as a potent β-secretase 1 allosteric inhibitor [169], regulating APP processing and decreasing Aβ production [170]. Moreover, sAPPα could affect APP processing also by inducing a significant down-regulation of *BACE1* expression (Figure 4d). Therefore, these data suggest that restoring sAPPα levels or enhancing its association with β-secretase 1 in AD patients to rebalance the impaired APP processing could be potential strategies to ameliorate AD symptoms, as also indicated by sAPPα inhibitory effects on Tau phosphorylation [171].

### 4.3. TSPAN3

The human *TSPAN3* gene is located on chromosome 15q24.3 and encodes for Tetraspanin-3 (previously known as Oligodendrocyte-Specific Protein (OSP)/Claudin-11-Associated Protein, OAP1), a N-terminally glycosylated transmembrane protein [172] belonging to the Tetraspanin superfamily (also named Tetraspans or Transmembrane 4 Superfamily, TM4SF), which comprises 33 mammalian members [173]. Tetraspanins are evolutionarily conserved, glycosylated and palmitoylated transmembrane proteins, characterised by four TMs (with TM1, 3 and 4 containing polar residues), a short extracellular loop (EC1), a very short (4 residues) intracellular loop, a long extracellular loop (EC2) subdivided into a constant region (containing the A, B and E α-helices) and a variable region, and short (8–21 residues) N-terminal and C-terminal cytoplasmic regions [173]. Structurally, all tetraspanins present the CCG motif after the B helix and two conserved cysteine residues forming intramolecular disulphide bonds. Cysteine number varies among different tetraspanins, with members presenting two additional cysteines to seven or eight cysteine residues, forming both intramolecular and intermolecular disulphide bonds [173]. In addition to interact with a variety of transmembrane proteins—e.g., integrins, cell adhesion proteins, cytokines and Growth Factor (GF) receptors, membrane-bound enzymes—different tetraspanins can interact with each other via their TM domains, EC2 loop and the surrounding lipid composition. Acting in concert, they form an interaction network called “Tetraspanin Web” or Tetraspanin-Enriched Microdomains (TEMs), localising different membrane-bound molecular players in segregated microdomains in a tissue-specific fashion [174,175,176,177]. Tetraspanins have been reported to take part in the regulation of multiple biological processes, including cell adhesion, migration and proliferation, as well as regulation of immune responses, infection, cancer progression and nervous system development [173,178]. However, their supramolecular organisation and function at synapses is still being investigated [179,180,181]. Tetraspanin-3 presents six cysteine residues in the EC2 loop and was reported to interact with Integrin β1 and Claudin-11 to regulate oligodendrocyte proliferation and migration [172,182]. In addition to oligodendrocytes, Tetraspanin-3 is highly expressed also in astrocytes and neurons [172], where it interacts with Sphingosine-1-Phosphate Receptor 2 (S1PR2), regulating its clustering with its ligand Nogo-A to control cell spreading and neurite outgrowth inhibition [183]. Importantly, Tetraspanin-3 has been found highly expressed in human AD brains and demonstrated to act in concert with other tetraspanins to stabilise APP, ADAM10 and the γ-secretase complex in both the cell membrane and the endocytic pathway [184]. Since *TSPAN3* expression is positively regulated by sAPPα treatment (Figure 4e), altogether these data support the notion that sAPPα could also exert neuroprotective effects promoting the APP non-amyloidogenic pathway. Accordingly, as mentioned, sAPPα also induces a significant down-regulation of BACE1 expression, thus affecting Aβ formation.

### 4.4. ISM2

The human *ISM2* gene is located on chromosome 14q24.3 and encodes for Isthmin 2 (also known as Thrombospondin and AMOP containing Isthmin-like 1, TAIL1), which is present in three different isoforms [185]. Isthmin 2 belongs to the isthmin family, a group of secreted proteins [186] characterized by an N-terminal SP, a central Thrombospondin-1 (TSR1) domain and C-terminal Adhesion-associated domain in MUC4 and Other Proteins (AMOP) domain, as well as multiple sites of C-mannosylation and N-glycosilation [187]. The two different human isthmin genes, *ISM1* and *ISM2*, encode for secreted proteins of 50 and 64 kDa respectively. Isthmin 1 structure and functions have been investigated in different contexts, reporting both angiogenic and anti-angiogenic activities [188,189,190] as well as a pleiotropic role as adipokine [186,191], while Isthmin 2 is still poorly characterised. Isthmin 2 structure suggests a possible angiogenic activity, since, besides Isthmin 1, the only other proteins in the human genome that feature a C-terminal AMOP domain—namely, Mucin 4 Cell Surface Associated (MUC4) and Sushi Domain Vontaining 2 (SUSD2)—are important elements in angiogenesis [185]. However, the presence of the central TSR-1 domain in Isthmin 2 structure may indicate that, like Isthmin 1, can be both pro-angiogenic and anti-angiogenic [185]. In this regard, our cellular in vitro model shows that sAPPα significantly up-regulated *ISM2* expression (Figure 4f) suggesting a possible role in neuronal context. Interestingly, *ISM2* is mainly expressed in the placenta and has been associated with preeclampsia and choriocarcinoma [185]. However, multiple transcriptomic analyses revealed that *ISM2* is also highly expressed in the brain, in particular in the nucleus accumbens (part of the basal ganglia) [192]. Within this context, it should be emphasised that Deep Brain Stimulation (DBS) of the nucleus accumbens itself has been observed to enhance learning and cognitive functions after TBI [193,194]. Interestingly, electromagnetic stimulation of SH-SY5Y neuroblastoma cells resulted in the increased expression of ADAM10 and an enhanced sAPPα release [195], suggesting the existence of a positive feedback loop and hinting at a possible correlation with Isthmin 2 putative, yet to be demonstrated neuronal roles of interest for TBI.

### 4.5. VEGFA

The human *VEGFA* gene is located on chromosome 6p21.1 and encodes for the Vascular Endothelial Growth Factor A (VEGF-A or VEGF, previously known as Vascular Permeability Factor, VPF), a secreted and glycosylated protein member of a family including VEGF-B, VEGF-C and VEGF-D (implicated in lymphangiogenesis regulation), the virally encoded VEGF-E, VEGF-F (snake venom VEGF), Endocrine Gland-derived VEGF (EG-VEGF) and the Placental Growth Factor (PLGF) [196]. VEGF plays a variety of different roles with key functions in regulating vasculogenesis and angiogenesis in both homeostatic and pathological contexts—promotes growth of the vascular endothelium, participates in differentiation mechanisms, increases permeability and molecule transport, supports anti-apoptotic processes and contributes to blood and lymphatic vessels development [196]. As a consequence of *VEGF* mRNA alternative splicing, VEGF is present in different isoforms, namely VEGF_121_ (highly diffusible), VEGF_145_, VEGF_148_, VEGF_165_ (the most frequently expressed isoform), VEGF_183_, VEGF_189_ (extracellular matrix-bound isoform) and VEGF_206_, and each isoform exhibits a differential heparin-binding ability [197]. VEGF family members can bind to three different receptors belonging to the tyrosine kinase receptor superfamily: VEGFR1 (Fms-like tyrosine kinase 1, Flt-1) and VEGFR2 (Kinase-insert Domain Receptor, KDR) are mainly expressed on vascular endothelial cells and are bound by VEGF-A/VEGF-B and VEGF-A/VEGF-E, respectively, while VEGFR3 (Flt-4) is expressed only on lymphatic endothelial cells and is therefore bound by VEGF-C and VEGF-D. VEGF-A heparin-binding isoforms and PLGF can also bind Neuropilin 1 (NRP-1) to increase their affinity for VEGFR2, while VEGF-C and VEGF-D can bind NRP-2 through a similar interaction involving VEGFR3 to regulate lymphangiogenesis [197]. From the N-terminus to the C-terminus, VEGFs present a SP sequence, the N-terminal portion, the dimerisation sites, VEGFR1 binding site, N-glycosylation site, VEGFR2 binding site, Plasmin Cleavage Sequence (PCS), multiple heparin-binding sites and the Neuropilin binding site. However, VEGFs and VEGF-Rs expression is not limited to endothelial cells, but they are also found in blood system cells, tumour cells and neurogenic cells [198]. VEGF has been shown to contribute to neuronal development and regeneration in the CNS [199,200] and to exert neuroprotective and neurorestorative effects [201]. In particular, VEGF has been recognised to promote neurogenesis, neuronal survival, proliferation and migration, as well as axonal growth and guidance. VEGF exerts its effects also in neuron-related cell types, by favouring glia survival, and glia and oligodendrocyte migration [202]. The modulation of VEGF expression has been suggested as a potential mechanism associated with AD development and its clinical deterioration [203] and VEGF was found lowly expressed in patients with AD [204]. In this regard, our preliminary in vitro observations showed that sAPPα could exert its neuroprotective effects through the regulation of VEGF expression (Figure 5a,b) specifically the pro-angiogenic isoforms VEGF_165_ and VEGF_189_ (Figure 5c). Within this context, as a consequence of alternative splicing, the 8-exon *VEGF* pre-mRNA is differentially spliced to form mRNAs encoding for the so-called VEGF_XXX_b isoforms [205]. The differential splice-acceptor-site selection in the 3′UTR within exon 8 results in the presence of two sub-exons, 8a and 8b, distinguishing an alternate family of VEGF termed VEGF_XXX_b (opposed to the canonical family VEGF_XXX_), mainly formed by VEGF_121_b, VEGF_145_b, VEGF_165_b, VEGF_183_b, VEGF_189_b and VEGF_206_b [205]. All VEGF_XXX_ family members feature the 8a sub-exon, while VEGF_XXX_b isoforms present the 8b sub-exon. The alternative C-terminal region allows VEGF_XXX_b family members to inhibit the pro-angiogenic, proliferative, migratory and vasodilatory properties of VEGF_XXX_ proteins [205,206]. The observed sAPPα-mediated up-regulating effects on *VEGF* mRNA were due to a sAPPα-induced increased stability of *VEGF* mRNA (Figure 5d) mirrored by an increased VEGF release (Figure 5e).

An additional up-stream player in the regulation of VEGF expression is the PKCβII/ELAV (Embryonic Lethal Abnormal Vision) cascade [207,208,209]. ELAV are a small family of proteins, which includes the ubiquitously expressed ELAVL1 (also known as HuA and HuR) and the neuron-specific members neuronal ELAV (nELAV, namely HuB, HuC and HuD). These RNA binding proteins act post-transcriptionally to dictate the fate of several transcripts during their journey from the nucleus to the cytoplasm [210]. At brain level, nELAV proteins contribute to regulate neuronal differentiation and maintenance, and are involved in synaptic plasticity associated with learning and memory processes [211,212,213,214]. The amount of nELAV proteins was found significantly reduced, along with clinical dementia progression, in the hippocampi from AD patients, where it also negatively correlated with Aβ levels [215]. Further, the direct treatment of human SH-SY5Y cells with both Aβ_40_ [216] and Aβ_42_ [215] induces a strong decrease in nELAV content together with a reduction in ADAM10, the best characterised α-secretase involved in APP non-amyloidogenic pathway and sAPPα production. Noteworthily, sAPPα treatment resulted in the increased translocation of nELAV in the nucleus (Figure 6a) and cytoskeleton (Figure 6c,d), along with their concomitant decrease in the cytosol (Figure 6b). Notably, the cytoskeleton represents an important site of protein synthesis and, at this level, a parallel rise in PKCβII amount was also observed (Figure 6c–e) which is in line with PI3K/Akt activation [141]. This strongly suggests the implication of the PKCβII/nELAV cascade in the regulation of VEGF expression at neuronal level as well.

The observed VEGF increase in human SH-SY5Y cells after sAPPα exposure is in line with data showing that VEGF is able to improve the cognitive decline in Tg2576 AD mouse model [217]. Besides the importance of this vascular factor in AD, Cerebrovascular Injury (CVI) is a recognised hallmark of TBI that affects function and integrity of the cerebrovascular system, contributing to neuronal dysfunction and neurodegeneration [218]. VEGF has been reported to mediate TBI amelioration [219], suggesting that new vasculogenic therapies targeting VEGF and its expression could be a feasible approach for vascular-related brain injuries typical of AD and TBI.

## 5. Discussion

The functions of APP and its cleavage fragments in the CNS have been extensively investigated in the past, especially for their relationship with AD. As a consequence of the increasingly aging population and the improvement of quality of life, the prevalence of AD and other neurodegenerative diseases is also increasing. In parallel, TBI has been effectively recognised as an important risk factor for the development of sporadic AD. Hence, it is of great significance to elucidate the role of APP and its cleavage peptides in these diseases. APP fragments produced through non-amyloidogenic pathways, particularly sAPPα, exert favourable effects in the CNS with important neuroprotective actions. Conversely, APP fragments generated via the amyloidogenic and non-canonical pathways have been demonstrated to have detrimental effects in the CNS. Indeed, amyloidogenic pathway-generated AICDs aggravate AD progression through positive feedback mechanisms by transcriptionally up-regulating *APP* and *BACE1* expression. Noteworthily, an increasing number of β-secretase 1 substrates has been reported together with the accumulation of neurotoxic APP fragments (e.g., η-CTF and Aη-α) upon β-secretase 1 genetic and pharmacological inhibition [64]. Consequently, specifically blocking β-secretase 1-mediated APP cleavage without affecting other substrates and ensuring that alternative neurotoxic APP fragments accumulation does not occur under the investigated therapeutic intervention will be mandatory for future research in this field. Therefore, pharmacological approaches aiming to reduce the production of detrimental APP fragments and, at the same time, to increase the generation of non-amyloidogenic pathways peptides could be feasible strategies for the prevention and treatment of AD, and may also provide new guidance for TBI treatment.

### 5.1. A Possible Therapeutic Intervention via sAPPα?

Since TBI has been proven as an important risk factor for AD, sAPPα modulatory activity on gene transcription may be of great interest for its neurotrophic effects. The ability of sAPPα not only to inhibit β-secretase 1 activity [169,170] but also to transcriptionally regulate *BACE1* mRNA expression may have important consequences on future β-secretase 1-targeting therapies aiming to reduce the production of amyloidogenic APP fragments. The sAPPα-mediated up-regulation of *TSPAN3* mRNA, together with its reported stabilising activity towards APP, ADAM10 and γ-secretase [184], suggest Tetraspanin-3 as a possible novel drug target to increase APP non-amyloidogenic processing. In addition, its importance for S1PR2/Nogo-A clustering and Nogo-A important role in AD pathogenesis due to its ability to modulate Aβ generation [220], further hints at Tetraspanin-3 as a possible important player in the sAPPα neuroprotective circuit. Although *ISM2* has not been investigated yet in neuronal context, the ability of sAPPα to mediate its up-regulation at mRNA level together with its elevated expression in the nucleus accumbens, important in TBI context [193,194], may suggest a possible role in sAPPα neuroprotective network and warrant further analyses. Finally, the early VEGF up-regulation and the late VEGF down-regulation mediated by sAPPα suggest a physiologic fine-tuning of its production in which isthmins may play a possible important role (although available data are limited to *ISM1* and cancer context [188,190]). Moreover, besides *VEGF*, *ADAM10* mRNA is also a target of PKC/nELAV pathway [216] and Aβ was shown to impair ADAM10 expression both in vitro and in AD hippocampi [215]. VEGF has been shown to improve the cognitive decline in Tg2576 AD mouse model by inducing *ADAM10* expression and decreasing *BACE1* production [217], thus suggesting the existence of a loop between APP cleavage fragments functions, PKC/nELAV/VEGF pathway and the potential protective functions of non-amyloidogenic sAPPα. These considerations may have possible important consequences in the dysregulated APP metabolism occurring in several conditions (e.g., from physiological aging and AD to brain injury response) [141] and point towards possible sAPPα-based therapeutic strategies. The neuroprotective, neurotrophic and neurogenic properties of sAPPα relevant against AD and TBI are mediated via the HBS located in APP E1 domain, within residues 96–110 [221]. In addition, the peptide APP derivative APP96–110 has been shown effective towards TBI following intravenous administration [222,223]. In this regard, recent studies both in vitro and in vivo reported that sAPPα acute administration in mice models or its chronic delivery through gene therapy effectively ameliorated both AD and TBI symptoms, further hinting at the proposed neurotrophic and neuroprotective roles of sAPPα [224,225].

### 5.2. A Putative Interaction Network for the Receptor for Activated C Kinase 1?

We previously demonstrated that sAPPα activates the PI3K/Akt/NF-κB pathway, influencing PKCβII signalling via the up-regulation of the Receptor for Activated C Kinase 1 (RACK1) [141]. RACK1 is a scaffold and ribosomal protein involved in a variety of molecular mechanisms, with crucial roles in both physiologic and pathologic conditions. RACK1 exerts its roles in several cellular contexts, including the immune system [226,227,228,229,230,231,232,233,234,235], cancer cells [236,237,238,239,240], intestinal homeostasis [241,242] and neurons [2,44,141,243,244,245]. In this regard, at a neuronal level, RACK1 has been reported to be required for point contact formation, axon outgrowth [246,247], dendritic arborisation [248], corticogenesis [249], synaptic plasticity, addiction, learning and memory by regulating N-methyl D-Aspartate (NMDA) receptor function [250], metabotropic glutamate receptor 1/5 (mGluR1/5)-triggered control of protein synthesis via its association with PKCβII [2], protection against oxidative stress [251] and BDNF expression [252,253,254]. Notably, reduced RACK1 levels were observed in both aged rat and human AD brains [255,256,257]. Aβ oligomers decrease RACK1 distribution in the membrane fraction of cortical neurons impairing PKC-mediated GABAergic transmission [258]. Moreover, RACK1 decreased levels have been also reported to induce Beclin-1-mediated autophagy in neurons [259]. In TBI context, RACK1 exerts neuroprotective effects via the activation of the Integrated Stress Response (ISR) pathway involving Inositol-Requiring Enzyme 1 (IRE1) and X-box Binding Protein 1 (XBP1) [260]. Although RACK1 has been initially discovered as the scaffold of different isoforms of activated PKC [231,245], literature data indicate that its interaction network is still increasing. Thanks to its β-propeller structure, RACK1 is able not only to interact with a broad range of binding partners, but also to participate in a variety of cellular mechanisms with potential implications also in neuronal context. Besides its well-known interaction with different PKC isoforms, RACK1 interacts directly or indirectly with genes/proteins here discussed or mentioned, including MAPK [261] and ADAM10 [262] via PKC signalling in neuronal context and VEGF via the PI3K/Akt/mTOR pathway [263,264,265,266], although these observations were limited to cancer cells. In addition, both RACK1 [2] and nELAV [267] are observed within stress granules, although their putative interaction has still to be demonstrated. Finally, despite the negative correlation between *ISM1*/Isthmin 1 and VEGF in cancer context [188,190], literature data correlating *ISM2*/Isthmin 2 with VEGF are still lacking, although a possible interaction could be hypothesised based on its pro- or anti-angiogenic properties. Therefore, the reported interactions among the different players presented in the previous sections and the possible existence of additional correlations—not yet demonstrated or investigated—indicate a possible important role played by VEGF and RACK1 in the construction of a sAPPα-related interaction network, with potential implications for neuroprotection (Figure 7).

## 6. Materials and Methods

### 6.1. Chemicals

Wortmannin (PubChem CID: 312145), sAPPα and the RNA polymerase II inhibitor 5,6-dichloro-1-beta-D-ribofuranosylbenzimidazole (DRB) (PubChem CID: 5894) were obtained from Sigma-Aldrich (St. Louis, MO, USA). Wortmannin and DRB were dissolved in DMSO at concentration of 100 mM and frozen in stock aliquots, then diluted at the concentration of use in culture medium. All reagents for cell culture were supplied by EuroClone (Milan, Italy). Mouse monoclonal anti-α-tubulin was purchased from Sigma-Aldrich. Rabbit polyclonal anti-PKCβII, rabbit polyclonal anti-VEGF and the anti-nELAV were obtained from Santa Cruz Biotechnology (Santa Cruz, CA, USA). Rabbit polyclonal anti-p42/44 MAPK and anti-phospho-p42/44 MAPK (Thr202/Tyr204) were purchased from Cell Signaling Technology, Inc. (Danvers, MA, USA). Host-specific peroxidase conjugated IgG secondary antibodies were purchased from Pierce (Rockford, IL, USA). Electrophoresis reagents were from Bio-Rad (Richmond, CA, USA).

### 6.2. Cell Culture and Treatments

Human neuroblastoma SH-SY5Y cells from the European Collection of Cell Cultures (ECACC No. 94030304) were cultured in 1:1 MEM-Ham’s F-12 medium mixture supplemented with 10% Foetal Bovine Serum (FBS), 2 mM glutamine, 100 U/mL penicillin, 100 μM streptomycin, 1% non-essential amino acids. Cells were maintained at 37 °C in a humidified 5% CO_2_ atmosphere. At 24 h before treatments, cells were cultured in medium without FBS and antibiotics. Subsequently, cells were treated with 10 nM sAPPα and/or Wortmannin 0.2 µM for different timings as reported in figure legends. When indicated, cells were exposed to 50 μM DRB [214] (or DMSO as vehicle control) for 1 h before sAPPα treatment.

### 6.3. Subcellular Fractionation

A total of 2 × 10^6^ cells were seeded in 60 mm dishes and incubated in fresh serum-free medium for 24 h at 37 °C before treatments. After treatment, cells were washed with 1 × PBS and homogenised 15 times using a Teflon glass homogeniser in fractionation buffer (0.32 M sucrose, 20 mM Tris-HCl pH 7.4, 2 mM EDTA, 10 mM EGTA, 50 mM β-mercaptoethanol, 0,3 mM phenylmethylsulfonyl difluoride, 20 μg/mL leupeptin). Homogenates were centrifuged at 3600× *g* for 5 min to the obtain the nuclear fraction; supernatants were centrifuged at 100,000× *g* for 30 min to separate cytosol and membrane fractions. The pelleted membrane fraction was sonicated in the same fractionation buffer supplemented with 0.2% (vol/vol) Triton X-100, incubated at 4 °C for 45 min and then centrifuged at 100,000× *g* for 30 min to separate membrane (supernatant) and cytoskeleton (pellet) fractions. Aliquots of the different fractions were assayed for protein quantification via the Bradford method.

### 6.4. Plasmid DNA Preparation, Transient Transfections and Luciferase Assays

Plasmids were purified with the HiSpeed^®^ Plasmid Midi Kit (Qiagen, Valencia, CA, USA). DNA was quantified and assayed for purity using a DUR24 530UV/Vis Spectrophotometer (Beckman Coulter Inc., Fullerton, CA, USA). Transient transfections were carried out using Lipofectamine^®^ 2000 (Invitrogen, Carlsbad, CA, USA) following manufacturer’s instructions and performed in 24 multi-well culture plates. For each well, 7 × 10^5^ cells were seeded in MEM-Ham’s F-12 medium without FBS, antibiotics and supplemented with 1% L-glutamine. The pGL3-VEGF luciferase-reporter construct plasmid (indicated as Δ1_VEGF and obtained as indicated in literature [268,269]) was co-transfected with pRL-TK Renilla luciferase expressing vector to measure transfection efficiency (Promega, Madison, WI, USA). During transfection, SH-SY5Y cells were incubated at 37 °C in 5% CO_2_ and then treated directly for with 10 nM sAPPα. Cells were then lysed with 1 × Passive Lysis Buffer provided by Dual-Luciferase Reporter Assay System according to manufacturer’s specifications (Promega, Madison, WI, USA). Luminescence was measured employing a 20/20n Luminometer (Turner BioSystems, Sunnyvale, CA, USA) with 10 s integration time.

### 6.5. qPCR

qPCR was carried out as previously described [270]. A total of 2 × 10^6^ cells were seeded in 60 mm dishes and treated as described. Total RNA was extracted using RNeasyPlus Mini Kit (Qiagen, Valencia, CA, USA) following manufacturer’s instructions. QuantiTect^®^ reverse transcription kit and QuantiTect^®^ SYBR Green PCR kit (Qiagen, Valencia, CA, USA) were used for cDNA synthesis and gene expression analysis following manufacturer’s specifications. QuantiTect^®^ primers for *BACE1*, *TSPAN3*, *ISM2*, *VEGFA* and *RPL6* were provided by Qiagen. Primers for *VEGFA* isoforms were purchased according to literature data [206]. *RPL6* was used as endogenous reference because it remained substantially stable in the 8 h time frame of the experiments with DRB [214]. Transcript quantification was performed via the 2^(−ΔΔCt)^ method.

### 6.6. mRNA Stability Analysis

The decay rate analysis of *VEGFA* mRNA was carried out as previously described [214]. 2 × 10^6^ cells were seeded in 60 mm dishes, pre-treated with 50 μM DRB for 1 h and then treated with 10 nM sAPPα. Cells were collected at different time points (0, 4, 6 or 8 h). Total RNA was extracted and reversed transcribed as previously mentioned. Kinetic determination of *VEGFA* mRNA levels was assessed via qPCR as previously described [214]. *VEGFA* mRNA levels were normalised to *RPL6* mRNA levels and expressed as a percentage of the initial steady-state *VEGFA* mRNA levels.

### 6.7. Immunoblot

Immunoblot samples were prepared by mixing the cell lysate with 5× sample buffer (125 mM Tris-HCl pH 6.8, 4% SDS, 20% glycerol, 6% β-mercaptoethanol, 0.1% bromo-phenol) and denaturing at 95 °C for 5 min. Samples were electrophoresed into a 10% SDS-polyacrylamide gel under reducing conditions. Proteins were transferred to poly-vinylidene fluoride (PVDF) membrane (Amersham, Little Chalfont, UK), blocked in TBS-Tween 5% non-fat dry milk, and subsequently incubated with primary antibodies (mouse anti-α-tubulin, anti-nELAV, anti-p42/44 and anti-phospho-p42/44 1:1000, rabbit anti-PKCβII 1:300, rabbit anti-VEGF 1:150) diluted in TBS-Tween 5% non-fat dry milk. Immuno-reactivity was measured using host-specific secondary IgG peroxidase conjugated antibodies (1:5000 diluted) and ECL.

### 6.8. Densitometry and Statistics

Following immunoblot image acquisition through an AGFA scanner and analysis by means of the NIH IMAGE 1.47 program (Wayne Rasband, NIH, Research Services Branch, NIMH, Bethesda, MD, USA), the relative densities of the bands were expressed as arbitrary units and normalised to data obtained from control samples run under the same conditions. Data were analysed using the analysis of variance test followed, when significant, by an appropriate post hoc comparison test. A *p* value < 0.05 was considered significant. The data reported are expressed as mean ± SEM of at least three independent experiments.

## 7. Conclusions

The identification of further APP proteolytic cleavages in addition to the canonical α-, β-, and γ-secretases considerably complicates the whole APP biological frame. This raises important questions not only on the physiological role of APP full-length and its cleavage fragments, but also their possible pathological contributions in both AD and TBI contexts. Evidence shows that most non-canonical APP secretases, particularly δ- and η-secretases, increase Aβ production and generate neurotoxic fragments. Hence, investigating their actual contribution to AD development and progression may suggest possible future pharmacologic strategies also for TBI. Further research is required to fully understand the complexity of APP biology in health and disease. Among the different APP fragments, sAPPα neuroprotective effects have been addressed and studied in different research models, both in vitro and in vivo, strengthening the correlation between sAPPα modulating effects on gene transcription and its observed neuroprotective effects. Indeed, investigating sAPPα ability to modulate the expression of genes relevant for neuroprotection in both AD and TBI contexts may offer relevant and feasible approaches for future pharmacologic interventions.

## Figures and Tables

**Figure 1 ijms-24-06639-f001:**
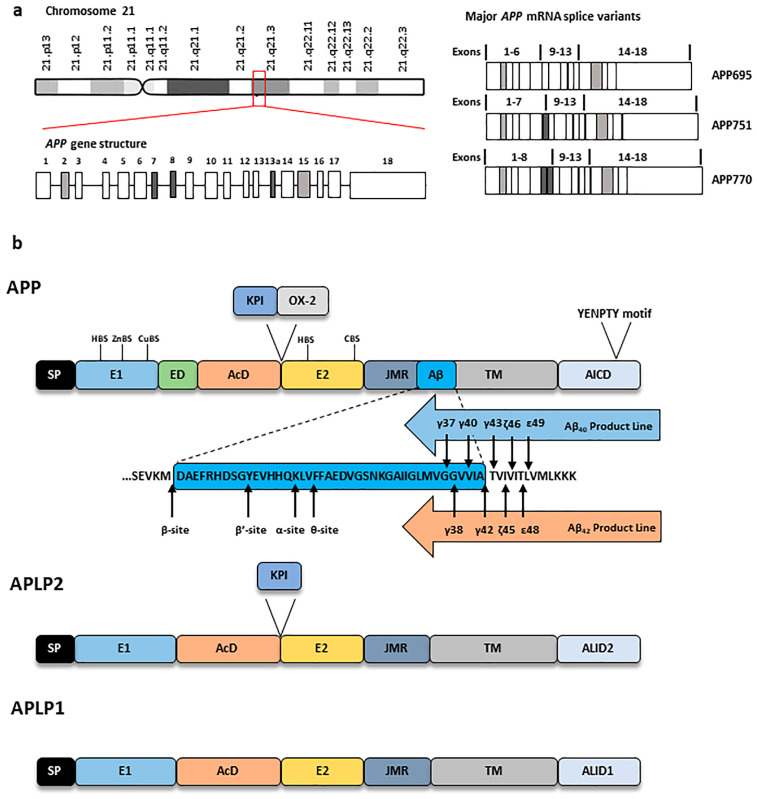
APP gene, mRNA and protein structure. (**a**) Structure of *APP* gene and mRNA. *APP* gene, located on chromosome 21q21.3, features 18 exons. Alternative splicing of exons 7 and 8 (*dark grey*) leads to the expression of APP695, 751 and 770 major isoforms, while differential splicing of exons 2 and 15 (*light grey*) generates APP639 and L-APP, respectively. (**b**) Structure of the three APP protein family members APP, APLP1 and APLP2. From the N-terminus to the C-terminus, APP features the cysteine-rich E1 domain (with Heparin-Binding Site (HBS), a Zinc-Binding Site (ZnBS) and Copper-Binding Site (CuBS)), the Extension Domain (ED), the Acidic Domain (AcD), the helix-rich E2 domain (with a second HBS and a Collagen-Binding Site (CBS)), the Juxtamembrane Region (JMR), the Aβ sequence, the Transmembrane Domain (TM) and APP Intracellular Domain (AICD) which contains a YENPTY sorting motif. APP751 and APP770 contain the additional Kunitz Protease Inhibitor (KPI) domain and an OX-2 antigen domain. Amino-acid sequence of Aβ region is shown along with the different secretases cleavage sites as well as the Aβ product lines. Both APLP1 and APLP2 lack Aβ sequence and present the APLP-intracellular domain 1 (ALID1) and ALID2, respectively. APLP2 features a KPI domain similarly to some APP isoforms.

**Figure 2 ijms-24-06639-f002:**
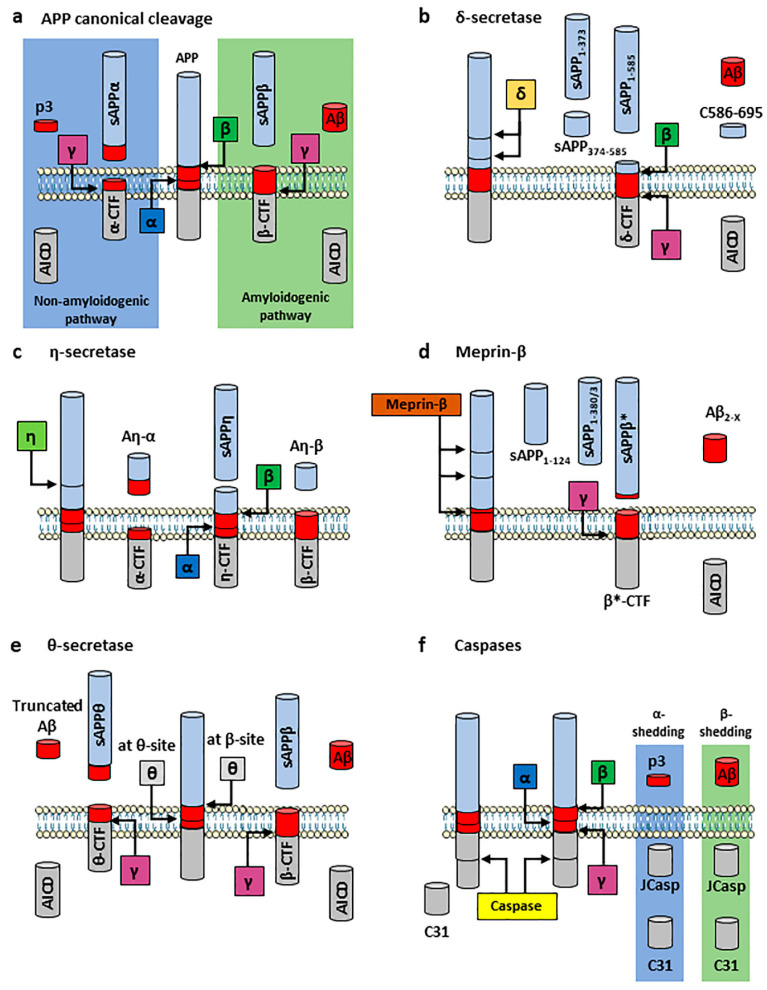
Schematic overview of APP-processing pathways. (**a**) APP canonical proteolytic processing. In the canonical cleavage, APP is either processed in the non-amyloidogenic pathway, where the sequential cleavage by α-secretase (ADAM10) (*blue square*) and the γ-secretase complex (*purple square*) produces sAPPα and p3, or in the amyloidogenic pathway, where the sequential cleavage by β-secretase 1 (*green square*) or Cathepsin B and the γ-secretase complex liberates sAPPβ and Aβ. While α-produced AICD are rapidly degraded in the cytoplasm, β-generated AICD form the transcriptional factor ATF complex together with Fe65 and Tip60, that translocates into the nucleus to up-regulate APP-related genes (see text for details). (**b**–**f**) APP non-canonical proteolytic processing. (**b**) APP processing mediated by δ-secretase (*orange square*) releases three soluble APP fragments (sAPP_1–585_, sAPP_1–373_, and sAPP_374–585_) and δ-CTF, which is further processed by β- and γ-secretases, releasing Aβ, AICD and the C586–695 fragment. (**c**) η-secretase (*light green square*) releases sAPPη and η-CTF, which is further processed by either α- or β-secretase 1, releasing Aη-α or Aη-β respectively; the remaining CTFs are cleaved by the γ-secretase complex, releasing p3 and AICD, or Aβ and AICD respectively. (**d**) Meprin-β (*brown rectangle*) produces sAPPβ* (similar to sAPPβ) and two shorter soluble fragments (sAPP_1–124_ and sAPP_1–380/3_), while the remaining β*-CTF is processed by the γ-secretase complex, releasing AICD and Aβ_2–X_. (**e**) θ-secretase (*grey square*) can either cleave APP at the θ-site, releasing sAPPθ and a θ-CTF that is further processed by the γ-secretase complex releasing AICD and a truncated form of Aβ, or act as a conditional β-secretase. (**f**) Caspase-3, -6 and -8 (*yellow rectangle*) cleave within APP Intracellular Domain, releasing C31 while, after the sequential cleavage operated by the γ-secretase complex, the small peptide JCasp is released in the cytoplasm.

**Figure 3 ijms-24-06639-f003:**
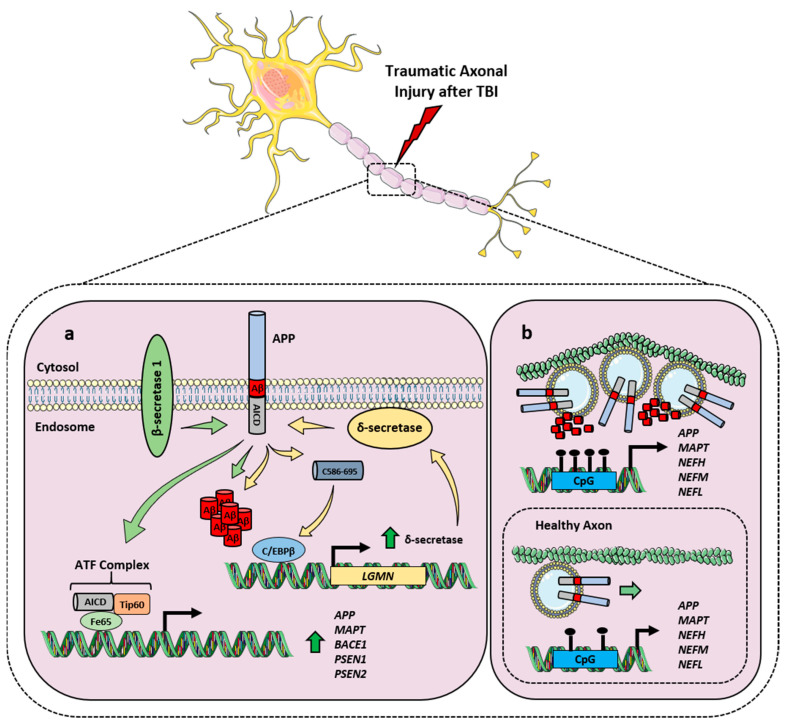
Alteration of APP processing after TBI. (**a**) Acute axonal damage after TBI induces the expression of APP, β-secretase 1 and γ-secretase components PS1 and PS2 (*dark green arrow*), as well as an increased Aβ generation and deposition. In addition, TBI triggers C/EBPβ activation, which induces the expression of δ-secretase (*dark green arrow*) that, in turn, cleaves APP forming Aβ and the C586–695 peptide that activates C/EBPβ. This suggests the establishment of a deleterious vicious loop possibly correlated to AD development. (**b**) Epigenetic and mechanical components of TBI-triggered APP processing alterations. After TBI, a differential CpG methylation of *APP*, *MAPT*, Neurofilament Heavy (*NEFH*), Neurofilament Medium (*NEFM*) and Neurofilament Light (*NEFL*) is observed [92]. Moreover, the induced axonal cytoskeleton distortion impairs the fast-axonal transport of APP (*light green arrow*), which accumulates in proximity of the axonal injury site, resulting in Aβ deposition.

**Figure 4 ijms-24-06639-f004:**
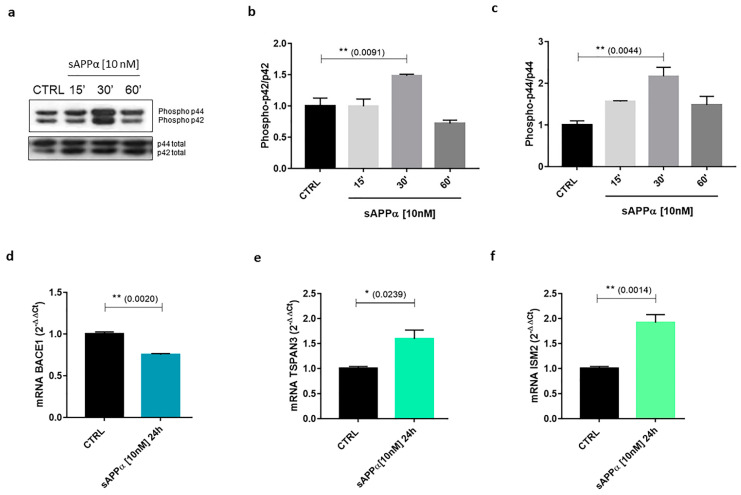
Effects of sAPPα treatment on early MAPK activation and *BACE1*, *TSPAN3* and *ISM2* transcriptional regulation. Human neuroblastoma SH-SY5Y cells were cultured and treated with 10 nM sAPPα for 15, 30 or 60 min (**a**–**c**) or for 24 h (**d**–**f**) as previously detailed [141]. Western blot analysis and qPCR were performed as previously reported [141]. (**a**–**c**) Evaluation of early MAPK activation. The image is a representative Western blot. Phosphorylation of p42 and p44 was normalised to their respective total p42 and p44 levels. (**d**–**f**) Evaluation of sAPPα-mediated transcriptional regulation on *BACE1*, *TSPAN3* and *ISM2*. mRNA levels were evaluated by qPCR (endogenous reference, *RPL6*). Results are expressed as mean ± SEM, n = 3 independent experiments. Statistical analysis was performed with one-way ANOVA followed by Dunnett’s multiple comparison test (**b**,**c**) or with Student’s *t*-test (**d**–**f**) with * *p* < 0.05 and ** *p* < 0.01 vs. control (CTRL). Statistical significance is detailed in the respective figure panel.

**Figure 5 ijms-24-06639-f005:**
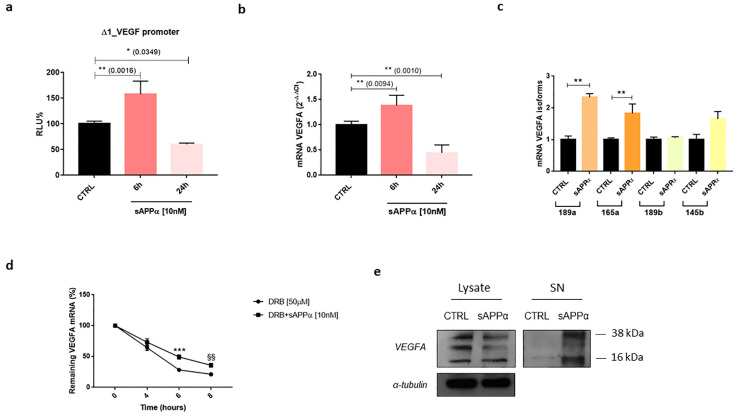
Effects of sAPPα treatment on VEGF expression. SH-SY5Y cells were treated with 10 nM sAPPα for 6 or 24 h. (**a**) SH-SY5Y cells transiently transfected with Δ1_VEGF luciferase-reporter plasmid and treated with sAPPα were lysed and luciferase activity was measured as previously described [141]. Luciferase activity is expressed as RLU% normalised to non-treated construct (set as 100%). (**b**,**c**) Evaluation of sAPPα-mediated *VEGFA* transcriptional regulation. mRNA levels were assessed by qPCR and normalised on *RPL6* (**b**) or the respective *VEGFA* isoform (employed primers described in [206]) (**c**). (**d**) Evaluation of sAPPα effects on *VEGFA* mRNA stability. SH-SY5Y cells were pre-treated with 50 µM DRB 53-85-0 (a classic RNA polymerase II inhibitor), then treated with 10 nM sAPPα for 0, 4, 6 or 8 h. RNA was extracted and reverse-transcribed as previously described [141]. mRNA levels were assessed by qPCR. Results are expressed as a percentage of the initial steady-state *VEGFA* mRNA levels. (**e**) Evaluation of VEGF protein levels in cell lysates and supernatants (SN). The image is a representative Western blot. Results are expressed as mean ± SEM, n = 3 independent experiments. Statistical analysis was performed with one-way ANOVA followed by Dunnett’s multiple comparison test (**a**–**c**), with Bonferroni multiple comparison test (**d**) or Student’s *t*-test (**e**) with * *p* < 0.05 and ** *p* < 0.01 vs. control (CTRL), *** *p* < 0.001 DRB vs. DRB + sAPPα (6 h) and §§ *p* < 0.01 DRB vs. DRB + sAPPα (8 h). Statistical significance is detailed in the respective figure panel.

**Figure 6 ijms-24-06639-f006:**
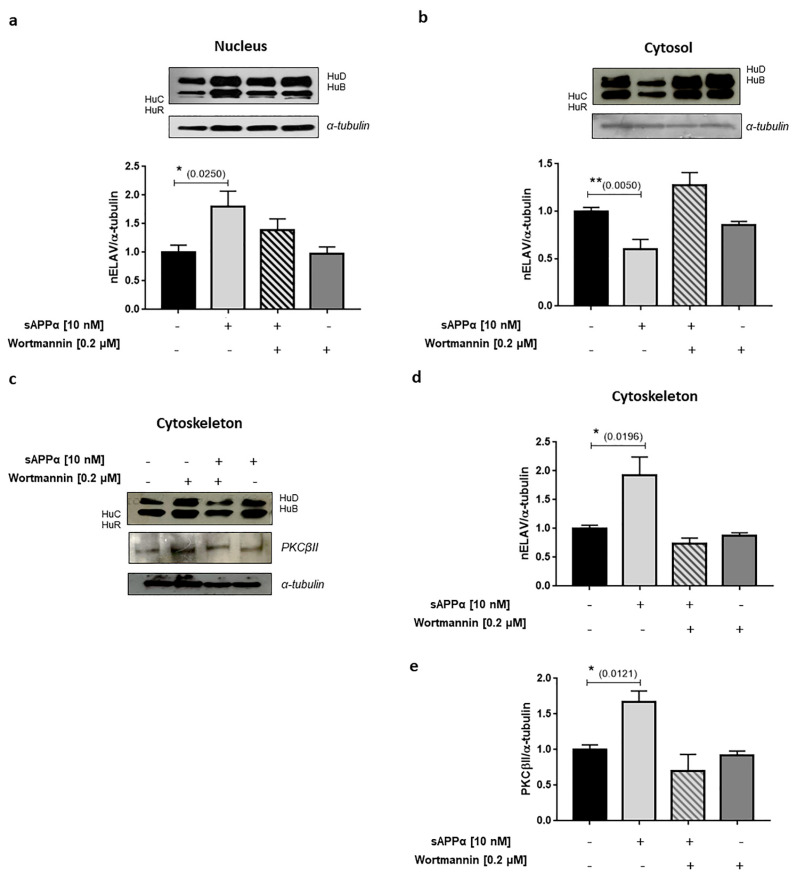
Evaluation of the PKCβII/nELAV/VEGF pathway in sAPPα-mediated effects on VEGF. (**a**–**e**) SH-SY5Y cells were treated with 10 nM sAPPα alone or in combination with 0.2 µM Wortmannin (an irreversible PI3K inhibitor). Subcellular fractionation was performed as previously described [141]. The image is a representative Western blot. nELAV protein levels were analysed in the nucleus (**a**), cytosol (**b**) and cytoskeleton (**c**,**d**) fractions. PKCβII protein levels were analysed in the cytoskeleton fraction (**c**,**e**). Protein levels were normalised to α-tubulin expression. Results are expressed as mean ± SEM, n = 3 independent experiments. Statistical analysis was performed with one-way ANOVA followed by Dunnett’s multiple comparison test with * *p* < 0.05 and ** *p* < 0.01 vs. control (untreated cells). Statistical significance is detailed in the respective figure panel.

**Figure 7 ijms-24-06639-f007:**
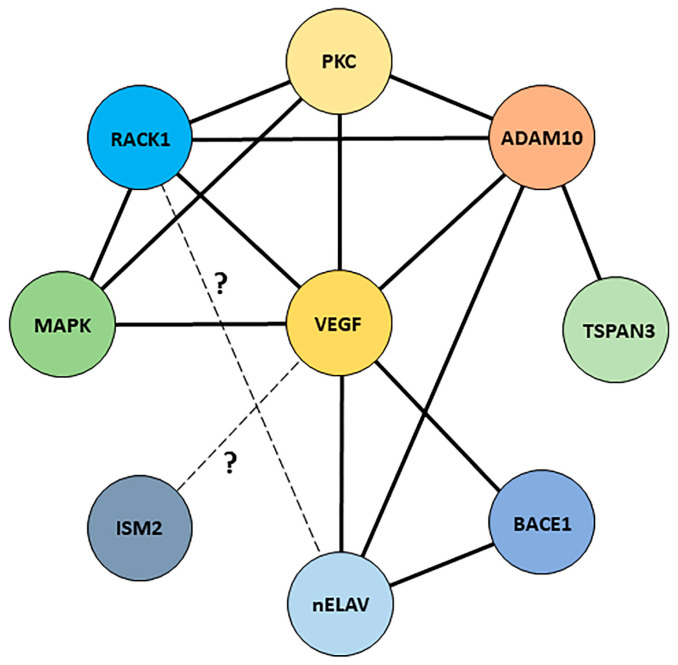
Putative sAPPα-related VEGF-RACK1 neuroprotective interaction network. RACK1 has been demonstrated to play pivotal roles in neuronal context and its modulation has been observed to be influenced by sAPPα treatment. Therefore, considering the reported and putative interactions among RACK1 and several players here discussed, it is possible to hypothesise a molecular network correlated to sAPPα and with potential implications in both AD and TBI context, from both a biological and pharmacological point of view (*bold lines* = protein-protein interactions or protein functional correlations reported in literature and cited in the text; *dash lines* = putative structural and/or functional protein correlations hypothesised based on available literature data).

**Table 1 ijms-24-06639-t001:** APP proteolytic fragments and their reported functions.

APP Processing	APP Fragment	Cell Line/Model	Characteristics and Functions	Ref.
-	APP full-length	PC12 pheochromocytoma cells; APP- B103 cells; embryonic carcinoma P19 and NT2 cell lines; NB-1 neuroblastoma cell line; SK-N-MC cells (human neuroblastoma cell line); rat hippocampal neurons; in vivo (hAPP751 or hAPP695 over-expressing mice, APP^−/−^ mice, Down’s Syndrome (DS) Ts65Dn and 1YeY mice)	Acting as cell surface receptor; involved in neuronal adhesion and iron transport; promotes cell division, neurite outgrowth, axonogenesis, synapse formation and maintenance and synaptic plasticity; neuroprotective role against Aβ and glutamate toxicity	[58]
**Canonical—Amyloidogenic**	sAPPβ	Human Embryonic Stem Cells (hESCs)	Induction of stem cells neural differentiation	[95]
		Rat hippocampal cells	100-fold less active neurotrophic effects compared to sAPPα in protecting hippocampal neurons against excitotoxicity, Aβ-induced toxicity, and glucose deprivation	[96]
		Sprague–Dawley rats brain slices	No marked changes in Long-Term Potentiation (LTP) compared to sAPPα	[97]
		N9 cells (myc-immortalized murine microglial cell line),	Stimulation of microglia activation through MAP kinase signaling pathways (i.e., ERKs, p38 kinase, JNKs) and NF-κB activity; production of proinflammatory and neurotoxic products (e.g., iNOS, IL-1β, ROS)	[98,99]
	N-APP	E13 rat dorsal spinal cord explant; mouse sensory and motor neuron explants; dissociated sensory neuron cultures; In vivo (DR6 KO, Bax KO and p75NTR KO mice)	Interaction with DR6 to recruit caspase-3 and caspase-6 (in cell bodies and axons respectively) and triggering of axonal pruning, but also neuronal death and AD development	[42,43]
	β-CTF (C99)	Neuro-2a cells (neuroblast cell line); in vivo (mouse)	LTP disruption	[100]
		In vivo (APP^−/−^ and APP^+/−^ mice)	Synaptotoxicity induction and spine density reduction	[101]
		In vivo (3xTgAD with PS1_M146V_, βAPP_swe_ and TauP_301L_ transgenes); APP695-H4 cells (human glioma cell line)	Early pathological accumulation and learning and memory deficits	[102,103]
		PC12 cells (pheochromocytoma cell line); SK-N-MC cells (human neuroblastoma cell line); rat neuronal cultures; transgenic mouse models	Selective neurotoxicity, cortical atrophy, loss of hippocampal granule cells, astrogliosis, Aβ and APP immunoreactivity, impaired working memory, neocortical and hippocampal neurodegeneration and gliosis	[104,105,106,107]
	β′-CTF (C89)	-	*Unknown physiologic and pathologic properties*	-
	Aβ	PC12 pheochromocytoma cells; APP- B103 cells; embryonic carcinoma P19 and NT2 cell lines; NB-1 neuroblastoma cell line; SK-N-MC cells (human neuroblastoma cell line); rat hippocampal neurons; in vivo (hAPP751 or hAPP695 over-expressing mice, APP^−/−^ mice, Down’s Syndrome (DS) Ts65Dn and 1YeY mice)	Major APP metabolic fragment involved in AD development and progression - Aβ_40_ reported to act as transcription factor; involved in cholesterol transport and kinase activation; promotes neurogenesis and neurite outgrowth; exerts neuroprotective effects against oxidative stress; inhibits Aβ_42_ oligomerisation; interacts with extracellular matrix components (e.g., laminin and fibronectin) to promote neurite proliferation. - Aβ_42_ inhibits LTP and synaptic transmission; enhances Long-Term Depression (LTD); reduces synaptic spine density; promotes synaptic injury and cognitive impairment	[58]
	AICD	SH-SY5Y cells wild type (wt) and APP/APP_swe_); Mouse Embryonic Fibroblasts (MEF) and mouse brains (wt, PS1/2^−/−^, APP/APLP2^−/−^ and APP_ΔCT15_)	Transcriptional regulation, together with Tip60 and Fe65 and forming the ATF complex, of APP and AD-related genes	[108,109]
		Differentiated PC12 cells; rat primary cortical neurons	Induction of neurotoxicity by up-regulating the expression of GSK-3β and activating p53	[110,111]
		AICD-transfected Jurkat cells	Apoptosis induction through Fas-Associated protein with Death Domain (FADD)-induced programmed cell death	[112]
**Canonical—Non-amyloidogenic**	sAPPα	SH-SY5Y cells; B103 cells; rat primary cortical neurons; murine hippocampal neurons; MEFs; in vivo (wt and APP^−/−^ mice)	Induction of Akt neuronal cell survival-correlated pathway, facilitation of normal neurophysiological functions (e.g., memory functions) and promotion of neurite outgrowth	[113,114,115]
		In vivo (wt, APP_swe_/PS1_ΔE9_ and APP^−/−^ mice)	Neuroprotective effect against synaptic dysfunction and TBI	[116,117]
		In vivo (APP^−/−^ and APLP2^−/−^ mice); Sprague–Dawley rats brain slices	Contribution to the regulation of synaptic plasticity and LTP induction	[97,118]
	α-CTF (C83)	In vivo (APP^−/−^ and APP^+/−^ mice)	Induction of synaptotoxicity and reduction of spine density	[101]
		CHO cells	Indirect promotion of survival by lowering C99 levels	[119]
		In vitro assay; HEK293-APP cells	Hypothesised to be a γ-secretase inhibitor	[120]
	p3	In vitro assay; APP^−/−^ MEFs; human cortical neurons	Induction of neuronal excitotoxicity by contributing to the formation of Ca^2+^-permeable ion channels	[121]
		AD brain human samples	Accumulation in amyloid plaques	[122]
		Differentiated THP-1 cells (human monocyte line); MG7 cells (microglia cell line); D30 (murine astrocyte line); U373 cells (human astrocyte line)	Induction of apoptosis, inflammatory responses and neurotoxic effects by producing proinflammatory cytokines (e.g., Interleukin (IL)-1α, IL-1β, IL-6, Tumour Necrosis Factor-α (TNF-α), chemokine MCP-1)	[123]
**Non-canonical—** **δ** **-secretase**	sAPP_1–373_ sAPP_1–585_ sAPP_374–585_	GFP-APP- and GST-APP-HEK293 cells; primary cultured neurons; in vivo (wt and AEP^−/−^ 5xFAD and APP/PS1 mice)	sAPP_1–373_, but not sAPP_1–585_ nor sAPP_374–585_, exhibited neurotoxic properties	[60]
	δ-CTF	AD brain human samples	Accumulation in brain lysates from AD patients	[60]
**Non-canonical—** **η** **-secretase**	sAPPη (sAPP95) η-CTF (C191)	Mouse hippocampal cultures; acute hippocampal slices; in vivo (Thy1-GCaMP6 mice)	Binding to the γ-Aminobutyric Acid (GABA) receptor and modulation of GABAergic neurotransmission	[124]
		In vivo (5xFAD MT5-MMP^−/−^ mice); AD brain human samples	Accumulation in the surrounding of amyloid plaques in dystrophic neurites of transgenic mice and AD patients and contribution to cognitive decline	[63,64]
		In vitro assay; SH-SY5Y/APP_751_ cells; H4/APP_751_ human neuroglioma cells	Processed by Cathepsin L, although its physiological role has not been investigated	[125]
	Aη-α Aη-β	Murine hippocampal neurons and brain slices; human CSF	Both detected in mouse brain homogenates and human CSF (5-fold higher levels than Aβ); Aη-α, but not Aη-β inhibited LTP, suppressed neuronal activity and resides in the halo of the amyloid plaque.	[64]
**Non-canonical—** **Meprin-β**	sAPP_1–380/3_ sAPP_1–124_	-	*Unknown physiologic and pathologic properties*	-
	CTF	-	*Unknown physiologic and pathologic properties*	-
	Aβ_2–X_	In vivo (APP_swe_-based mouse models); AD patient brain samples	Increased potential to aggregate compared to Aβ_1–X_ peptides	[126]
	p11	Differentiated SH-SY5Y cells; in vivo (wt, APP^–/–^ and APLP2^–/–^ C57BL6J × 129/Sv mice); AD brain human samples	Increased production detected during neuronal differentiation	[127]
**Non-canonical—** **θ-secretase** **Non-canonical—Caspases**	sAPPθ	-	*Unknown physiologic and pathologic properties*	-
	θ-CTF (C80)	-	*Unknown physiologic and pathologic properties*	-
	APP-Ncas	-	*Unknown physiologic and pathologic properties*	-
	APP-Ccas (C31)	Neuro-2a cells; AD brain human samples	Synaptic dysfunction, neuronal apoptosis and death; APP-dependent toxicity	[128]
	JCasp	Primary neuronal cultures	Induction of neuronal apoptosis by transducing a tyrosine-dependent signalling	[129]
**Non-canonical—Cathepsin B**	Aβ_3–X_ Aβ_11–X_ (pE-Aβ)	In vitro assay; HEK293, 20E2, 2EB2, HAW and BAW cell lines; human AD brain samples	High aggregation propensity, cellular toxicity, disruption of LTP and proposed as predominant Aβ peptide species in AD brain patients	[38]
**Other secretases**	Aβ_4–X_ Aβ_5–X_	In vivo (APP_swe_ mouse models); AD patient brain samples	Increased potential to aggregate compared to Aβ_1–X_ peptides	[126]
	N-APP_18–286_	neonatal C57BL/6 × SJL mice hippocampal cultures	Binding an unknown neuronal receptor and increase of phosphatidylinositol phosphate levels	[77]
	APP_1–119_ APP_1–121_ APP_1–122_ APP_1–123_ APP_1–126_	Human CSF	~12 kDa sAPP fragments reported by mass spectrometry analyses with unknown physiologic or pathologic properties	[76]
	17–28 kDa APP N-Terminal Fragments	SH-SY5Y cells with HSV-1-mediated APP expression; rat cortical neuronal cultures	Generated through a developmentally regulated PKC-related APP processing pathway independent of α-secretase or β-secretase 1, with a putative important role in APP physiological function	[127,130]

**Table 2 ijms-24-06639-t002:** In vitro sAPPα-mediated differentially regulated genes obtained from the differential display analysis performed on SH-SY5Y neuroblastoma cells (*green highlight* = up-regulated genes of interest; *orange highlight* = down-regulated genes of interest).

Gene	Accession Number	Location	Encoded Protein	sAPPα-Mediated Regulation
** *TSPAN3* **	**NM_198902.3**	**15q24.3**	**Tetraspanin-3**	**Up-regulated**
** *ISM2* **	**NM_199296**	**14q24.3**	**Isthmin 2**	**Up-regulated**
*MNAT1*	AY165512.1	14q23.1	MNAT1 component of CDK activating kinase	Up-regulated
*HSPA8*	NM_153201.4	11q24.1	Heat Shock Protein Family A (Hsp70) Member 8	Up-regulated
*SOX11*	AB028641.1	2p25.2	SRY-Box Transcription Factor 11	Up-regulated
*USE1*	AB097050.1	19p13.11	Unconventional SNARE in the ER 1	Up-regulated
*KALRN*	NM_007064.5	3q21.1-q21.2	Kalirin RhoGEF kinase	Up-regulated
*MAP3K5*	NM_005923	6q23.3	Mitogen-Activated Protein Kinase Kinase Kinase 5	Up-regulated
*PTGES3*	NM_006601.7	12q13.3; 12	Prostaglandin E Synthase 3	Up-regulated
** *BACE1* **	**BC065492.1**	**11q23.3**	**β** **-secretase 1**	**Down-regulated**
** *VEGFA* **	**AF022375.1**	**6p21.1**	**Vascular Endothelial Growth Factor A**	**Down-regulated**
*RBM8A*	NM_005105.5	1q21.1	RNA Binding Motif Protein 8A	Down-regulated
*SLC27A3*	BC003654.2	1q21.3	Solute Carrier Family 27 Member 3	Down-regulated
*METTL21A*	BC009462.1	2q33.3	Methyltransferase 21A, HSPA Lysine	Down-regulated
*RPS28*	NM_001031.5	19p13.2	Ribosomal Protein S28	Down-regulated
*RPL24*	NM_000986.4	3q12.3	Ribosomal Protein L24	Down-regulated
*UBE3A*	NM_130839.5	15q11.2	Ubiquitin Protein Ligase E3A	Down-regulated
*QDPR*	BC000576.2	4p15.32	Quinoid Dihydropteridine Reductase	Down-regulated
*MTX1*	BC001906.1	1q22	Metaxin 1	Down-regulated
*COMMD1*	NM_152516.4	2p15	Copper Metabolism Domain Containing 1	Down-regulated
*PSMC6*	NM_002806.5	14q22.1	Proteasome 26S Subunit, ATPase 6	Down-regulated

**Table 3 ijms-24-06639-t003:** Reported involvement in AD and TBI of sAPPα-correlated up- and down-regulated genes here not investigated.

Gene	Reported Correlation with AD/TBI	Ref.
*HSPA8*	Decreased in AD patients’ samples and possible molecular biomarker for prognosis among HSP70 family in AD; part of the alcohol-sensitive protein networks modulated by AD-associated proteins that exacerbate neural and behavioural pathology upon alcohol drinking; potential drug target for preventing protein misfolding aggregation and cell death in AD and other neurodegenerative pathologies	[146,147,148]
*SOX11*	Hypothesised to participate in the modulation of neuron plasticity in the dentate gyrus of the hippocampus; involved in the early attempts of axon regeneration and neuronal survival; modulates peripheral nerve regeneration in adult mice and BDNF expression in an exon promoter-specific manner; favours endogenous neurogenesis and locomotor recovery in mice spinal cord injury	[149,150,151,152,153]
*USE1*	Involved in ER morphology maintenance and in the regulation of ER stress-induced neuronal apoptosis	[154]
*KALRN*	Key role in synaptic plasticity and in dendritic arbours and spines formation; proposed as possible therapeutic target for pharmacological intervention for synapse dysregulation; its deficit contributes to AD-related cognitive decline and memory loss; prevents dendritic spine dysgenesis induced by Aβ oligomers	[155,156,157]
*MAP3K5*	Neuroprotective action against apoptosis as part of the Notch1/ASK1/p38 MAPK signalling pathway; neuroprotective role against oxidative stress; involved in neuronal death after its IRE1α-mediated recruitment upon ER stress and UPR	[158,159,160]
*RBM8A*	Potential contribution to AD pathophysiological changes by regulating components of key complexes in autophagy	[161]
*SLC27A3*	Highly expressed in in human neural stem cells and involved in early brain development	[162]
*RPS28*	Promotion of P-body assembly and hypothesised to be linked with neurodegeneration	[163,164]
*UBE3A*	Involved in synaptic function and plasticity; age-dependently decreased in AD mice models; possible critical player in AD pathogenesis and potential therapeutic target	[165,166]
*COMMD1*	Hypothesised to be involved in hypoxia-induced AD neurodegeneration and brain injury via its interaction with Hypoxia-Inducible Factor 1 alpha (HIF-1α)	[167]

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
