# Peer review of "The Labyrinthine Landscape of APP Processing: State of the Art and Possible Novel Soluble APP-Related Molecular Players in Traumatic Brain Injury and Neurodegeneration"

_ijms, 2023, doi:10.3390/ijms24076639_

Round 1

Reviewer 1 Report

The manuscript titled "The labyrinthine landscape of APP processing: state of the art and possible novel soluble APP-related molecular players in traumatic braininjury and neurodegeneration" is well written and presented.

Please address the following:

1. For Figures 4, 5, and 6 please increase the image sizes and decrease white space.

2. Figure 7 is not self explanatory, so either change it to show what is described in the text and table 1 or delete this figure.

3. Have gridlines included for Table 1, for easier read.

Author Response

The manuscript titled "The labyrinthine landscape of APP processing: state of the art and possible novel soluble APP-related molecular players in traumatic brain injury and neurodegeneration" is well written and presented.

Please address the following:

  1. For Figures 4, 5, and 6 please increase the image sizes and decrease white space.

We thank the reviewer for the suggestion and modified the images’ layout accordingly.

  1. Figure 7 is not self-explanatory, so either change it to show what is described in the text and table 1 or delete this figure.

The reviewer is right in pointing out the issues regarding Figure 7. Indeed, we noticed that some errata in Figure 7 legend remained in the submitted manuscript, compromising its comprehension. Therefore, since we believe that Figure 7 can greatly summarise the take-home message of the “5.2. A putative interaction network for the Receptor for Activated C Kinase 1 (RACK1)?” section, we corrected the figure legend and included additional explanation to increase the image message delivery.

We modified the text as follows (lines 817-824): “RACK1 has been demonstrated to play pivotal roles in neuronal context and its modulation has been observed to be influenced by sAPPα treatment. Therefore, considering the reported and putative interactions among RACK1 and several players here discussed, it is possible to hypothesise a molecular network correlated to sAPPα and with potential implications in both AD and TBI context, from both a biological and pharmacological point of view (bold lines = protein-protein interactions or protein functional correlations reported in literature and cited in the text; dash lines = putative structural and/or functional protein correlations hypothesised based on available literature data).”

  1. Have gridlines included for Table 1, for easier read.

Although we completely understand the reviewer’s point, we would like to note that Tables 1, 2 and 3 layouts were directly managed by the journal editing staff and follow IJMS guidelines available in the manuscript template provided. Therefore, we prefer to leave the decision on whether edit the tables layout to IJMS editors and staff.

Reviewer 2 Report

In this hypothesis, the authors claims the state of the art and potential new soluble APP-related molecular drivers in traumatic brain injury and neurodegeneration: The tangled web of APP processing. Overall, this is an interesting hypothesis, but I feel that several lines have been copied directly from previous studies.

Please explain, what is the novelty of your hypothesis? There are many others sentences directly copied from other publications. Thus, I stopped my review and suggest to remove plagiarism from your manuscript first and resubmit this work for consideration in this journal.

Minor comments:

-Tables and figure legends needs to be revised.

-The English language also needs to be improved throughout the manuscript. The authors are advised to seek the assistance of a native English speaking colleague.

-I strongly encourage the authors to dedicate the time and improve the quality of this draft.

Author Response

In this hypothesis, the authors claim the state of the art and potential new soluble APP-related molecular drivers in traumatic brain injury and neurodegeneration: The tangled web of APP processing. Overall, this is an interesting hypothesis, but I feel that several lines have been copied directly from previous studies.

  • Please explain, what is the novelty of your hypothesis?

Based on our research interests, we produced this manuscript with two main aims: (1) comprehensively review the state of the art of APP processing to gather the up-dated information of APP cleavage fragments in physiologic and pathologic contexts; (2) focus our attention on the sAPPα fragment and the possible sAPPα-mediated regulation of genes of interest from an AD and TBI point of view. Therefore, in the present work, we hypothesised that, due to its neuroprotective and neurotrophic roles observed both in vitro and in vivo, sAPPα treatment on an in vitro pre-clinical model (i.e. SH-SY5Y cells) could result in the modulation of genes of interest for the pathologies here considered and pave the way to consider the existence of a possible correlation between Alzheimer's disease and TBI. In this regard, aside from the notion that sAPPα acts as a potent BACE1 allosteric inhibitor [Peters-Libeu et al. J Alzheimers Dis. 2015;47(3):545-55. doi: 10.3233/JAD-150282], no previously published data on the sAPPα-related gene targets we selected (i.e. BACE1, TSPAN3, ISM2, VEGFA) were available in literature. Therefore, the novelty of our hypothesis does not lie in the previously addressed sAPPα neuroprotective role, but rather in the targets that sAPPα is able to modulate via the MAPK cascade, with potential implications from a pharmacological point of view. In addition, considering the additional PKC/nELAV cascade and the role of RACK1 as molecular hub added a higher level of complexity to our hypothesis, allowing a novel, previously not considered molecular network in AD and TBI with a possible important pharmacologic impact. Hence, the purpose of our work was to critically discuss not-yet-considered molecular players to suggest and route future research on the field.

  • There are many others sentences directly copied from other publications. Thus, I stopped my review and suggest to remove plagiarism from your manuscript first and resubmit this work for consideration in this journal.

Considering the reviewer’s concerns regarding the supposed plagiarism, we submitted our manuscript to the software Compilatio for a professional plagiarism check. According to the report we obtained (that we attach in this response), a 27% plagiarism was detected throughout the whole manuscript. However, aside from the phrases detected in the Front Matter and in the Back Matter – which are strictly functional data and would naturally result in a very high percentage similarity – a total of 7 sentences (highlighted phrases n. 10-14, 17, 20 in the report) were found to be similar to previously published material. Therefore, we have re-written these selected phrases and highlighted in the text. Detected sentences n. 15, 16, 18, 19, 21-28 (highlighted in the report) were excluded as well since they are part of the Materials and Methods section and for their naturally intrinsic similarity to the standard scientific methodology.

We modified the text accordingly and highlighted the phrases:

  • Lines 43-44: “Neuropathologic markers like NFTs and senile plaques composed of Aβ aggregates hystopathologically characterise the brain tissue of AD patients”;
  • Lines 212-214: “This produces an externally released, N-terminally truncated APP form termed soluble APPβ (sAPPβ) and the membrane-associated C-terminal fragment (C99 or β-CTF)”;
  • Lines 230-231: “and the membrane-associated C-terminal fragment (C83 or α-CTF)”;
  • Line 276: “produces sAPPα and p3, or in the amyloidogenic pathway”;
  • Table 1: “Binding an unknown neuronal receptor and increase of phosphatidylinositol phosphate levels”;
  • Table 3: “favours endogenous neurogenesis and locomotor recovery in mice spinal cord injury”;
  • Line 783-785: “RACK1 is a scaffold and ribosomal protein involved in a variety of molecular mechanisms, with crucial roles in both physiologic and pathologic conditions. RACK1 exerts its roles in several cellular contexts”;

Minor comments:

  • The English language also needs to be improved throughout the manuscript. The authors are advised to seek the assistance of a native English-speaking colleague.

We agree with the reviewer on this point. To ensure a better comprehension of our work, we were assisted in the manuscript proof-reading for English language improvement by a colleague of ours and Italian Institute of Technology member, who is an English native speaker. Language proof-reading was performed in revision mode to highlight corrections from the previous form.

Reviewer 3 Report

Dear Authors,

the manuscript 'The labyrinthine landscape of APP processing: state of the art and possible novel soluble APP-related molecular players in traumatic brain injury and neurodegeneration' by Masi et al, is a comprehensive and exhaustivereview of the state of the art of APP proteolytic processing. You also added preliminary experiments to prove your hypothesis, an added value to the manuscript. The paper is well written, clear and sounds of interest for the scientific community.

Minor: revision of English

Author Response

Dear Authors,

the manuscript 'The labyrinthine landscape of APP processing: state of the art and possible novel soluble APP-related molecular players in traumatic brain injury and neurodegeneration' by Masi et al, is a comprehensive and exhaustive review of the state of the art of APP proteolytic processing. You also added preliminary experiments to prove your hypothesis, an added value to the manuscript. The paper is well written, clear and sounds of interest for the scientific community.

Minor: revision of English

We thank the reviewer for the kind words on the quality of our manuscript and for the upraising comments on our work here presented. We agree with the reviewer on the need for an English manuscript check to ensure the best comprehension. Therefore, we were assisted in the manuscript proof-reading by a colleague of ours and Italian Institute of Technology member, who is an English native speaker. Language proof-reading was performed in revision mode to highlight corrections from the previous form.

Round 2

Reviewer 2 Report

Authors did not revise carefully.